# SatDreamer360: Multiview-Consistent Generation of Ground-Level Scenes from Satellite Imagery

**Xianghui Ze[1], Beiyi Zhu[2], Zhenbo Song[1], Jianfeng Lu[1], Yujiao Shi[2]***

[1]Nanjing University of Science and Technology, [2]ShanghaiTech University
{zexh,songzb,lujf}@njust.edu.cn,{zhuby2025,shiyj2}@shanghaitech.edu.cn

## Abstract

Generating multiview-consistent $360°$ ground-level scenes from satellite imagery is a challenging task with broad applications in simulation, autonomous navigation, and digital twin cities. Existing approaches primarily focus on synthesizing individual ground-view panoramas, often relying on auxiliary inputs like height maps or handcrafted projections, and struggle to produce multiview consistent sequences. In this paper, we propose SatDreamer360, a framework that generates geometrically consistent multi-view ground-level panoramas from a single satellite image, given a predefined pose trajectory. To address the large viewpoint discrepancy between ground and satellite images, we adopt a triplane representation to encode scene features and design a ray-based pixel attention mechanism that retrieves view-specific features from the triplane. To maintain multi-frame consistency, we introduce a panoramic epipolar-constrained attention module that aligns features across frames based on known relative poses. To support the evaluation, we introduce VIGOR++, a large-scale dataset for generating multi-view ground panoramas from a satellite image, by augmenting the original VIGOR dataset with more ground-view images and their pose annotations. Experiments show that SatDreamer360 outperforms existing methods in both satellite-to-ground alignment and multiview consistency.

## 1 Introduction

Generating ground-level scenes from satellite imagery has attracted significant attention due to the broad coverage and low acquisition cost of satellite images. This task shows promising applications in autonomous driving (Villalonga Pineda (2021); Lu et al. (2024)), 3D reconstruction (Liu et al. (2024); Yan et al. (2024)) and data augmentation (Yang et al. (2023); Gao et al. (2023)) for downstream tasks. Many existing works (Li et al. (2024a); Lin et al. (2024); Xu & Qin (2024); Ze et al. (2025)) focus on generating individual ground images from satellite views, leaving the continuity of multi-ground views largely unaddressed. In this paper, we aim to synthesize multiple ground-view images from a single satellite image, controlled by a predefined trajectory. This introduces new challenges in maintaining both geometric consistency with the top-down satellite image and multiview coherence across the sequence of generated frames.

Early approaches (Isola et al. (2017a); Regmi & Borji (2018); Shi et al. (2022); Lu et al. (2020); Qian et al. (2023)) formulate cross-view synthesis as a one-to-one mapping problem, often implemented with Conditional Generative Adversarial Networks (cGANs). These methods focus on aligning representations at pixel or perceptual level. However, the extreme viewpoint disparity between top-down satellite views and street-level images leads to limited field-of-view overlap. Satellite images inherently miss key elements such as building facades, tree trunks, and other occluded details, making the ground view generation task highly under-constrained and naturally one-to-many.

Recent advances leverage latent diffusion models (LDMs) (Rombach et al. (2022)) to better handle this uncertainty (Li et al. (2024a); Lin et al. (2024); Deng et al. (2024); Xu & Qin (2024); Ze et al. (2025)). These methods introduce probabilistic modeling to produce diverse and high-fidelity ground

---

*Corresponding author

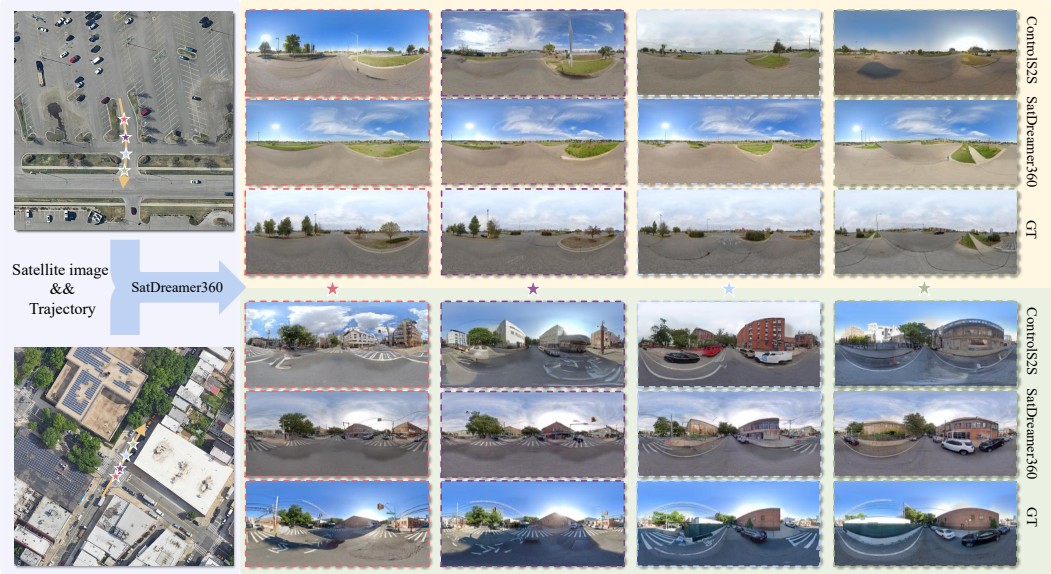

Figure 1: Given a satellite image and a sequence of query poses (colored stars), our goal is to synthesize coherent panoramic views along the trajectory. The proposed SatDreamer360 generates more realistic and geometrically consistent ground-level scenes compared to state-of-the-art methods, faithfully capturing spatial layouts and structural continuity across diverse environments.

images. However, they often rely on approximate projections (Lin et al. (2024); Ze et al. (2025)) or auxiliary data such as height maps (Li et al. (2024a); Deng et al. (2024); Xu & Qin (2024)), which can be difficult to obtain at scale. Moreover, while effective for single-view generation, these models fall short in producing multiview consistent sequences, which are critical for applications like simulation, planning, or digital twin city modeling. A recent effort (Xu & Qin (2025)) attempts to generate continuous ground-view videos by leveraging multi-angle satellite imagery in a two-stage pipeline: the first stage generates a base frame, followed by autoregressive generation of future frames. While this improves continuity, the reliance on multi-view satellite input and complex coordination for different generation stages reduces practical applicability.

In this paper, we present SatDreamer360, a unified framework that generates continuous and coherent ground-view sequences from a single satellite image and a target trajectory, as shown in Figure 1. The key idea is to embed explicit cross-view geometric reasoning between satellite and ground views, as well as across ground frames into the latent diffusion process.

We adopt a compact triplane representation (Huang et al. (2023); Bhattarai et al. (2024); Shue et al. (2023)) to encode scene geometry directly from the satellite image, avoiding the need of height maps (Deng et al. (2024); Xu & Qin (2025)) or handcrafted projections (Ze et al. (2025)). We further design a ray-based pixel attention mechanism that retrieves view-dependent features from the triplane and integrates them into conditional diffusion, enabling geometry-aware and controllable generation.

To enhance mutiview consistency, we draw inspiration from the use of epipolar constraints in pinhole cameras (Tobin et al. (2019); He et al. (2020); Huang et al. (2022; 2024)) and extend the idea to panoramic images with equirectangular projections. We design an epipolar-constrained attention module for panoramic images, which aligns features across frames by leveraging the known relative camera poses.

Finally, to support large-scale evaluation, we construct VIGOR++, an extension of the VIGOR dataset with ground-truth trajectories and continuous ground-view sequences, providing a new benchmark for cross-view generation. To summarize, our contributions are as follows:

- A unified framework, SatDreamer360, for generating continuous and geometrically consistent ground-view sequences from a single satellite image and a target trajectory.
- A ray-guided cross-view feature condition mechanism that encodes the 3D scene with a triplane representation and aggregates spatial cues pixel by pixel via ray-based attention, enabling geometry-aware and controllable diffusion-based ground-view generation.

- An interframe attention module that uses panoramic epipolar constraints via equirectangular projections to align features across frames, enhancing multiview consistency.
- A new VIGOR++ dataset, which extends VIGOR with continuous sequences and trajectory annotations, providing a benchmark for cross-view sequence synthesis.

## 2  RELATED WORK

**Cross-view ground scene generation** aims to reconstruct ground scenes from other perspectives, such as aerial (Xu et al. (2023); Gao et al. (2024)) or landmark images (Yang et al. (2023); Li et al. (2024b); Gao et al. (2023); Swerdlow et al. (2024)). Given the wide availability of satellite imagery, related research (Shi et al. (2022); Qian et al. (2023); Xu & Qin (2024); Ze et al. (2025); Li et al. (2021; 2024d); Xu & Qin (2025)) focuses on satellite-to-ground generation. Previous works (Isola et al. (2017a); Regmi & Borji (2018)) implicitly convert satellite image features into ground map representations, often causing geometric distortions. Later methods introduced approximate projections (Lu et al. (2020); Shi et al. (2022); Lin et al. (2024); Ze et al. (2025)), height maps (Marí et al. (2022); Lu et al. (2020); Deng et al. (2024); Xu & Qin (2024); Li et al. (2024a); Xu & Qin (2025)), or estimating density maps (Qian et al. (2023)) as priors. However, their accuracy is constrained by errors in the estimation priors, and ground-image-based methods often over-rely on these inaccurate projections while neglecting the broader contextual information available from satellite imagery.

**Multiview consistent image generation** aims to generate multiview continuous frames from given prompts. Early Gan-based approach (Vondrick et al. (2016); Rematas et al. (2022)) has been surpassed by methods utilizing the diffusion architecture (Blattmann et al. (2023b;a)), where Video Diffusion Models (VDM) (Blattmann et al. (2023b); Singer et al. (2022); Wu et al. (2023)) introduce spatiotemporal modules into U-Net to generate coherent sequences, though with high computational cost. Some methods designed for single-object multi-view generation, such as MVDream (Shi et al. (2023b)) and Zero123++ (Shi et al. (2023a)), implicitly encode camera embeddings directly into the diffusion process. However, when applied to large-scale scene generation, such implicit conditioning is insufficient to enforce strict pose constraints. Recent works (Tseng et al. (2023); Huang et al. (2024)) incorporate epipolar constraints to enforce multiview consistency, but are mostly limited to pinhole cameras, with little exploration of panoramic settings. More broadly, methods such as (Liu et al. (2023); Kong et al. (2024); Voleti et al. (2024); Bourigault & Bourigault (2024)) generate multi-view images from single-object inputs, yet they focus on object-level generation and cannot address scene-level continuity.

## 3  METHOD

Given a satellite image $S$ and a set of 4-DoF ground camera poses $\{p^i = [t^i, \psi^i]\}$, where $t^i$ denotes spatial location and $\psi^i$ the yaw angle, our goal is to synthesize a sequence of ground panoramic images $G^i$ that are spatially aligned with the satellite view and consistent across multiple views.

To obtain the optimal solution for ground image inference, as illustrated in Figure 2, we develop SatDreamer360 based on a latent diffusion model (Song et al. (2020); Blattmann et al. (2023b)) to synthesize ground-level views conditioned on the satellite image and camera pose. It generates ground images by iteratively denoising a random Gaussian noise for $T$ steps, learning to predict the Gaussian noise $\epsilon$ injected at each step $t$:

$$\mathcal{L} = \mathbb{E}_{z_0, c, \epsilon, t} \left[ \| \epsilon - \epsilon_\theta (\sqrt{\bar{\alpha}_t} z_0 + \sqrt{1 - \bar{\alpha}_t} \epsilon, t, c) \|^2 \right]. \tag{1}$$

Here, $\epsilon_\theta$ is the denoising network using U-Net, $c$ is the conditioning input—comprising the satellite image $S$ and pose $p^i$, $\bar{\alpha}_t$ is the variance schedule, and $\epsilon$ is drawn from a standard Gaussian distribution. Ground images $G$ are encoded using a VQ-VAE (Esser et al. (2021)) encoder $\mathcal{E}(G)$ to obtain latent codes $z$. For clarity, we refer to ground representations in latent space also as $G$ in what follows.

### 3.1  RAY-GUIDED CROSS-VIEW FEATURE CONDITIONING

**Spatial Representation via Triplanes.** To represent the 3D scene covered by the satellite image, we adopt a tri-plane structure (Chan et al. (2022); Huang et al. (2023)), a lightweight and expressive

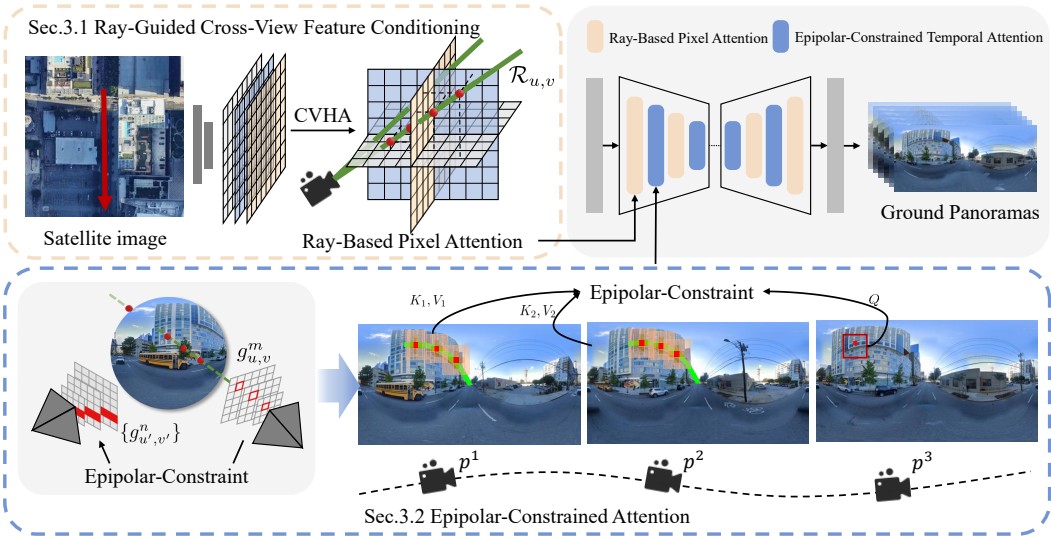

Figure 2: Overview of the proposed SatDreamer360 framework. Given a single satellite image and a target trajectory, our model synthesizes continuous ground-level panoramas along the path. A Ray-Based Pixel Attention mechanism retrieves view-specific features through cross-view geometric reasoning, guided by a tri-plane representation of the scene. An Epipolar-Constrained Attention module aligns features across frames using relative camera poses.

alternative, instead of the information-sparse BEV representation (Li et al. (2024c)) or the computationally intensive voxel representation (Li et al. (2022)). Three orthogonal planes $(XY, XZ, YZ)$ are defined in the tri-plane representation, with the $XY$ plane parallel to the ground.

Given a point in 3D space, its feature $F_{xyz}$ is obtained by aggregating the features from its projections onto the three planes:

$$F_{xyz} = F_{xy} \oplus F_{xz} \oplus F_{yz}, \qquad (2)$$

where $F_{xy}$, $F_{xz}$, and $F_{yz}$ denote the interpolated features from the corresponding 2D planes, and $\oplus$ denotes their element-wise summation to obtain the final 3D feature.

To construct the triplane representation, we initialize the planes by extracting features from the satellite image using a ResNet (He et al. (2016)), which naturally aligns with the top-down $XY$ plane. To enrich spatial reasoning across all three orthogonal planes, we apply Cross-view Hybrid Attention (CVHA Li et al. (2024c)), enabling interactions among the $XY$, $XZ$, and $YZ$ planes. Each plane aggregates projections from the other two, enriching its features with complementary spatial context. For instance, the updated features on the $XY$ plane are computed as:

$$F_{xy}^{\text{top}} = \text{CVHA}\left(F_{xy}^{\text{top}}, \text{Ref}_{xy}^{3D}\right), \quad \text{Ref}_{xy}^{3D} = F_{xy}^{\text{top}} \cup \{F_{yz_i}^{\text{side}}\} \cup \{F_{xz_i}^{\text{front}}\}. \qquad (3)$$

Here, $F_{xy}^{\text{top}}$ denotes the point feature on the $XY$ plane. The reference set $\text{Ref}_{xy}^{3D}$ contains local neighbors sampled along the $Z$-axis from the orthogonal $XZ$ and $YZ$ planes, denoted as $\{F_{xz_i}^{front}\}$ and $\{F_{yz_i}^{side}\}$. This cross-plane aggregation enables each point on the triplane to incorporate multi-view cues, thereby enhancing 3D spatial consistency. Moreover, in sequential settings, previously synthesized ground-view images can be projected back and integrated into the triplane to refine its representation. With CVHA, this incremental update yields a more expressive and temporally coherent scene model. Further architectural and implementation details are provided in the Appendix A.5.

**Ray-Based Pixel Attention.** Conventional cross-attention mechanisms (Rombach et al. (2022)) typically align global prompts with image-level semantics but often fail to respect underlying 3D scene geometry. This limits their ability to establish accurate cross-view correspondences, particularly in view synthesis tasks. To address this, we propose a Ray-Based Pixel Attention module that incorporates geometric priors by explicitly conditioning attention on camera rays.

Specifically, as illustrated in Figure 2 (top middle) and Appendix A.2, each pixel $g_{u,v}$ at location $(u, v)$ in the panoramic ground-view image $G \in \mathbb{R}^{H \times W \times C}$ corresponds to a unique 3D ray $\mathcal{R}_{u,v}$,

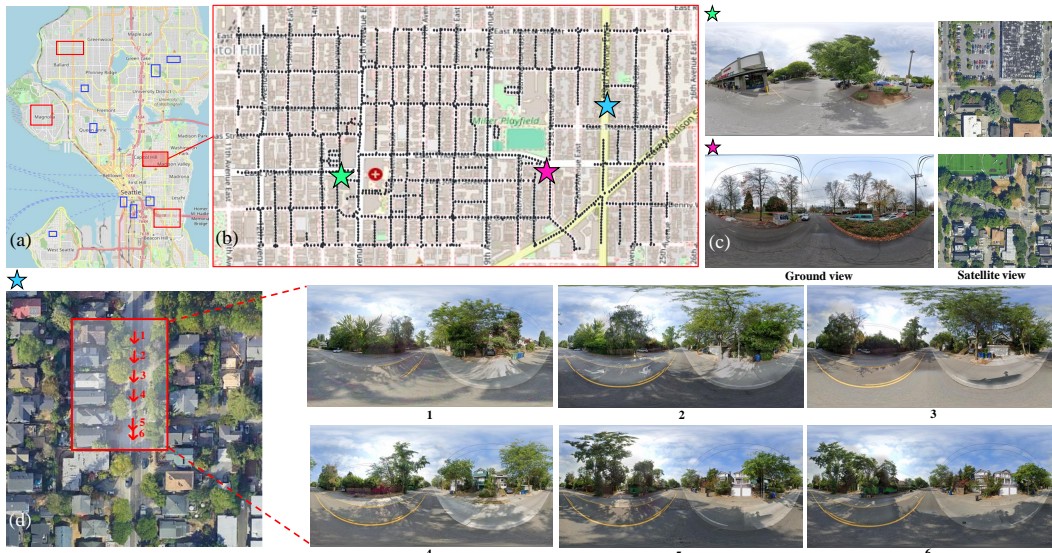

Figure 3: Overview of the VIGOR++ dataset. (a) The map of Seattle, USA, serves as an example of the ten cities in the dataset. The red boxes and blue boxes represent the districts for the training set and test set, respectively. (b) shows a road map. Dots and stars along the road represent locations of ground images and satellite images. Two of them, marked with the red star and green star, are shown in (c). (d) shows the continuous ground sequence within one satellite image.

parameterized by yaw $\psi$ and pitch $\theta$ angles:

$$\psi_{u,v} = (u - \frac{W}{2})/W \times 2\pi, \quad \theta_{u,v} = (\frac{H}{2} - v)/H \times \pi. \tag{4}$$

These angular parameters define the direction of the ray $\mathcal{R}_{u,v}$ in the camera coordinate system. The ray originates from the ground-view camera position, and its direction is uniquely defined by $(\psi_{u,v}, \theta_{u,v})$. To encode spatial cues along each ray, we sample $K$ points at evenly spaced depths $\{r_k\}_{k=1}^{K}$, and project them into the spatial coordinate system using the camera pose, yielding 3D positions $\mathbf{x}_{u,v,k}$. Features at these 3D positions are then extracted from the tri-plane representation using deformable attention:

$$F_{g_{(u,v)}} = \sum_{j=1}^{J} W_j \sum_{k=1}^{K} A_{k,j} \cdot F_{(\mathbf{x}_{u,v,k} + \Delta \mathbf{x}_{k,j})}, \tag{5}$$

where $J$ is the number of attention heads, $W_j$ is a learnable weight for head $j$, $\Delta \mathbf{x}_{k,j}$ is an offset along the ray around the sampled points, initialized to zero, and $A_{k,j}$ denotes the attention weight for these sampled points, normalized such that $\sum_{k=1}^{K} A_{k,j} = 1$ for each head. Both offsets and attention weights are dynamically refined across iterations, guided by the evolving ground latent feature map. $F_{(\mathbf{x}_{u,v,k} + \Delta \mathbf{x}_{k,j})}$ denotes features extracted from the triplane at the adjusted 3D positions using Eq. 2. The aggregated feature $F_{g_{(u,v)}}$ at pixel $(u,v)$ guides the U-Net in integrating satellite information, effectively aggregating spatial cues pixel by pixel.

## 3.2 EPIPOLAR-CONSTRAINED ATTENTION

To maintain mutiview consistency across consecutive frames in a lightweight and efficient manner, we draw inspiration from epipolar-constrained attention in pinhole images (Tseng et al. (2023)) and extend it to outdoor panoramic imagery. Specifically, we introduce an attention mechanism tailored for equirectangular projections. For two frames of ground panoramas, $G^m$ and $G^n$, a pixel $g_{u,v}^m$ on $G^m$ corresponds to a set of candidate pixels on $G^n$, enforced by the underlying geometric constraint (proof in Appendix A.3):

$$\left(P^{-1}(g_{u',v'}^n)\right)^{\top} \hat{t}_{mn} R_{mn} \left(P^{-1}(g_{u,v}^m)\right) = 0. \tag{6}$$

Here, the point set $\{g^n_{u',v'}\}$ on $G^n$ denotes candidate matches that satisfy the constraint relationships. The terms $R_{mn}$ and $t_{mn}$ denote the relative rotation and translation between frames $m$ and $n$, and $\hat{t}_{mn}$ is the skew-symmetric matrix of $t_{mn}$. The mapping $P$ is the equirectangular camera projection defined in Eq. 4. Therefore, when establishing mutiview consistency, we do not need to perform pixel-wise correspondence for the entire image as in previous work (Wu et al. (2023); Xu & Qin (2025)). Instead, we restrict attention to points that satisfy the epipolar constraint, significantly reducing redundancy while preserving geometric fidelity:

$$F_{g^m_{u,v}} = \mathrm{softmax}\left(\frac{QK^\top}{\sqrt{d}}\right)V, Q = W^Q F_{g^m_{u,v}}, K = W^K F_{\{g^n_{u',v'}\}}, V = W^V F_{\{g^n_{u',v'}\}}, \quad (7)$$

where $W^Q$, $W^K$, and $W^V$ are learnable matrices for query, key, and value. This epipolar-constrained attention is applied at multiple U-Net levels to fuse coarse and fine-grained features.

By restricting attention to points that satisfy the epipolar constraint, the computational complexity is reduced from $O(NHW \times NHW)$ to $O(NHW \times NM)$, where $N$ is the number of frames in sequences, $H$ and $W$ denote the height and width of the feature map, and $M$ is the number of sampled points satisfying epipolar constraints with $M \ll HW$. To further improve efficiency, we adopt a sparse querying strategy that uses only two reference frames: the very first frame of the sequence and the immediately preceding frame. The first frame acts as a global anchor, ensuring consistency in overarching scene characteristics such as weather and illumination. Simultaneously, the constraint with the previous frame preserves local geometric coherence.

### 3.3 VIGOR++: Extending VIGOR for Satellite-to-Ground Video Generation

Existing cross-view datasets lack continuous panoramic sequences. To address this, we construct VIGOR++, an extension of the VIGOR dataset (Zhu et al. (2021)) tailored for large-scale, consistent cross-view generation, enabling the dataset to be more widely used in 3D scene reconstruction, cross-view video localization tasks, as shown in Figure 3. To broaden the coverage of satellite maps for the task of large-scale scene generation, we expand the wide-area satellite map dataset by increasing it from the original $70\,\mathrm{m} \times 70\,\mathrm{m}$ to $160\,\mathrm{m} \times 160\,\mathrm{m}$ from Google Maps (goo (a)). Subsequently, we include additional cities. Apart from the initial cities of Chicago, New York, San Francisco, and Seattle, we integrate datasets for six additional regions: Atlanta, Bismarck, Kansas, Nashville, Orlando, and Phoenix. This augmentation enriches the variety of urban representations within the dataset.

To obtain continuous ground sequences, we extract all available Google Street View (goo (b)) images within the satellite region. Subsequently, we employed a semi-automatic approach to organize sampling paths for each satellite image. By leveraging sky color histograms and image embedding similarities, we constructed a connectivity graph and executed path extraction based on depth-first search to identify potential routes. Subsequent manual refinement ensured multiview coherence.

Our efforts yielded more than 90,000 novel cross-view satellite and ground video pairs. As shown in Figure 11, these images are evenly distributed across ten cities, covering a total area of 117.47 km². Most trajectories contain between 7 and 16 ground-view frames. The average frame interval is approximately 11 m, with the minimum interval being 0.079 m and the maximum reaching 20 m. Of these, 84,055 pairs are designated for training, while 7,443 are allocated for testing. To evaluate the model's generalization capabilities, the testing set is collected from locations entirely distinct from the training data.

## 4 Experiments

**Experimental Setup.** We use $256 \times 256$ satellite images and the 4-DoF camera poses of ground-view images as input, aiming to generate continuous ground-view sequences at a resolution of $128 \times 512$ for fair comparison with prior work. Our model is finetuned based on the pre-trained Stable Diffusion 1.5 model (Rombach et al. (2022)). In our experiments, we set the default number of ray samples to $K = 8$, and the number of points sampled along the epipolar lines to $M = 4$. Detailed analysis is provided in Appendix A.14. We first perform 300 epochs of finetuning on a single-image generation task, followed by an additional 300 epochs on continuous sequence data dataset to learn temporal consistency. During inference, we adopt DDIM sampling with 50 steps for efficient generation. All experiments are conducted using four NVIDIA L40 GPUs.

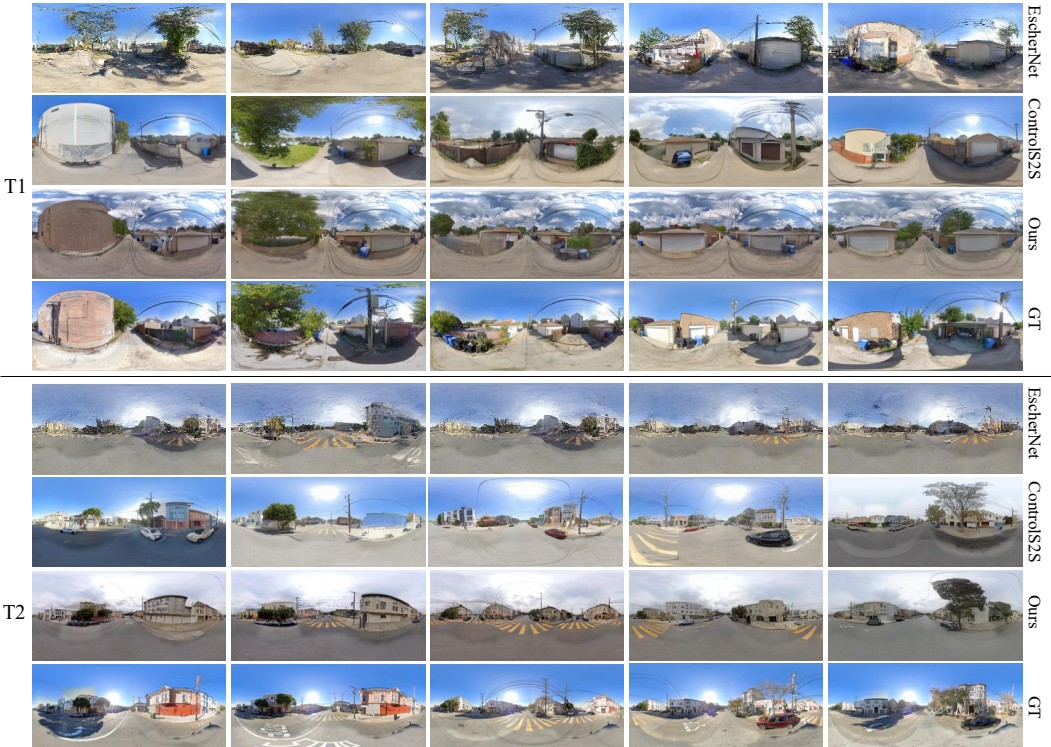

Figure 4: Qualitative comparison of ground-level image sequences along trajectories T1 and T2, shown from left to right. The corresponding satellite images and trajectories are provided in Figure 13. Our method produces more realistic textures and preserves structural and spatial continuity across frames, demonstrating stronger multiview coherence and environmental fidelity across diverse scenes.

Table 1: Quantitative comparison with existing algorithms on VIGOR++ dataset.

| Method | Perceptual level | | Semantic level | | Pixel level | | Multiview level | | ↓Depth |
|---|---|---|---|---|---|---|---|---|---|
| | ↓$P_{alex}$ | ↓FID | ↓DINO | ↓SegAny | ↑SSIM | ↑PSNR | ↓FVD | ↓CLIPSIM | |
| Sat2Den | 0.4584 | 133.6 | 4.437 | 0.3729 | 0.3892 | 12.06 | 11.70 | 8.405 | 7.671 |
| EscherNet | 0.5581 | 84.21 | 4.942 | 0.3845 | 0.2587 | 11.23 | 7.282 | 8.250 | 10.50 |
| ControlS2S | 0.4433 | 29.48 | 4.567 | 0.3753 | 0.3718 | 11.84 | 4.871 | 10.81 | 6.651 |
| Ours | **0.3955** | **27.41** | **4.156** | **0.3563** | **0.3964** | **12.75** | **2.101** | **6.820** | **5.623** |

**Datasets.** For the single ground-view image generation task, we use the CVUSA (Zhai et al. (2017)) dataset, which primarily focuses on rural areas, and VIGOR (Zhu et al. (2021); Lentsch et al. (2022)) dataset, which covers four major cities, following the same protocol as prior works (Shi et al. (2022); Qian et al. (2023); Ze et al. (2025)). These cross-view datasets provide one-to-one correspondences between panoramic ground images and satellite images. CVUSA contains 35,532 pairs for training and 8,884 pairs for testing, with most scenes focusing on rural areas. VIGOR comprises data collected from four cities—New York, Seattle, San Francisco, and Chicago—resulting in 52,609 pairs for training and 52,605 pairs for testing. For the continuous scene generation task, we conduct experiments using our proposed VIGOR++ dataset, where the training and testing sets are collected from entirely distinct regions.

**Evaluation Metrics.** We evaluate the authenticity and multiview consistency of generated images. For authenticity, we compare results with ground truth (GT) using pixel-level metrics (SSIM, PSNR, SD) and perceptual metrics based on pretrained networks ($P_{alex}$ (Krizhevsky et al. (2012)), $P_{squeeze}$ (Iandola et al. (2016)), and FID (Heusel et al. (2017))). Since real-world variations in weather and season can cause color shifts, strict pixel-level comparisons may be less informative. Following Ze et al. (2025), we emphasize structural and semantic similarity. We employ DINO (Caron et al. (2021)) and Segment Anything Kirillov et al. (2023) to extract semantic features, and use DepthAnything (Yang et al. (2024)) for depth consistency. In addition, we employ LRCE (Shen et al. (2022)) to measure the continuity of panoramic images along the left and right boundaries.

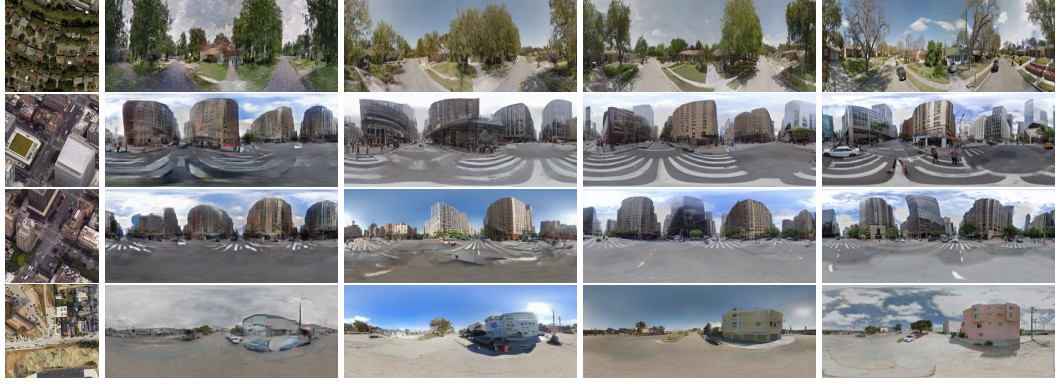

| (a) Sat | (b) Sat2Den | (c) ControlS2S | (d) Ours | (e) GT |

Figure 5: Qualitative comparison with previous works on satellite to single ground image generation, our model can effectively capture roadways, ground markings, and architectural details.

Table 2: Quantitative comparison with previous works on satellite to single ground image generation.

| | Method | Perceptual Level | | | Semantic Level | | Pixel Level | | | ↓Depth | ↓LRCE |
|---|---|---|---|---|---|---|---|---|---|---|---|
| | | ↓$P_{squeeze}$ | ↓$P_{alex}$ | ↓FID | ↓DINO | ↓SegAny | ↑SSIM | ↑PSNR | ↑SD | | |
| CVUSA | Pix2Pix | 0.3468 | 0.5084 | 44.51 | 5.242 | 0.3847 | 0.3190 | 13.20 | 12.08 | 21.85 | 18.81 |
| | S2S | 0.3218 | 0.4830 | 29.49 | 5.112 | 0.3852 | 0.3508 | 13.40 | 12.30 | 21.05 | 19.10 |
| | Sat2Density | 0.3217 | 0.4634 | 47.85 | 4.945 | 0.3763 | 0.3307 | 13.46 | 12.27 | 19.83 | 16.17 |
| | CrossDiff | - | - | 23.67 | - | - | 0.3710 | 12.00 | - | - | - |
| | ControlS2S | 0.3192 | 0.4323 | 21.30 | **4.807** | 0.3612 | 0.3753 | 13.67 | 12.33 | 19.58 | 14.62 |
| | Ours | **0.3146** | **0.4255** | **17.00** | **4.807** | 0.3602 | **0.3812** | **13.88** | **12.42** | **19.36** | **14.53** |
| VIGOR | Pix2Pix | 0.3346 | 0.4513 | 67.96 | 4.717 | 0.3833 | 0.3714 | 13.33 | 12.93 | 8.647 | 6.569 |
| | S2S | 0.3694 | 0.4941 | 121.1 | 5.032 | 0.4037 | 0.3273 | 12.16 | 12.31 | 10.87 | 9.790 |
| | Sat2Density | 0.2828 | 0.3898 | 54.49 | 4.408 | 0.3627 | 0.3956 | **14.14** | 12.38 | 8.054 | 5.805 |
| | ControlNet | 0.3395 | 0.4594 | 23.68 | 4.950 | 0.3916 | 0.3397 | 12.02 | 12.59 | 10.02 | 7.499 |
| | ControlS2S | 0.2729 | 0.3770 | 28.01 | 4.335 | 0.3529 | 0.4228 | 13.80 | 13.07 | 7.095 | 5.176 |
| | Ours | **0.2598** | **0.3469** | **21.36** | **4.287** | **0.3471** | **0.4385** | 14.08 | **13.11** | **6.727** | **5.081** |

Multiview consistency across frames is evaluated using Fréchet Video Distance (FVD) (Unterthiner et al. (2018)) and CLIP-based similarity (CLIPSIM) (Wu et al. (2021)), two common metrics for assessing temporal coherence and perceptual alignment in generated video sequences. FVD quantifies the distributional similarity between generated and real videos by computing the Fréchet distance between feature representations extracted from a pretrained video recognition network, capturing both spatial appearance and temporal dynamics. In addition, CLIPSIM measures semantic and perceptual similarity using representations from the CLIP model, which encodes visual content in a joint vision–language embedding space. By comparing these embeddings, CLIPSIM estimates how well semantic information is preserved across frames or between generated and reference videos.

## 4.1 COMPARISON WITH PRIOR WORK ON SATELLITE-TO-GROUND SEQUENCE GENERATION

Generating continuous and coherent ground-level Sequences from a single satellite image is highly challenging due to the extreme viewpoint gap and inherent spatial ambiguity. We compare our method against three representative baselines: Sat2Density (Qian et al. (2023)) and ControlS2S (Ze et al. (2025)), both designed for cross-view image generation, and EscherNet (Kong et al. (2024)), a recent diffusion-based model for general multiview synthesis.

Since neither the code nor data of Deng et al. (2024); Xu & Qin (2025) is released, comparisons with them cannot be conducted and thus are not included. StreetScape (Deng et al. (2024)) relies on ground-truth height maps, which indeed simplifies the problem but also creates a strong dependency on data that is often unavailable or costly to acquire. Sat2GroundScape (Xu & Qin (2025)) requires multiple satellite views to reconstruct a 3D model and uses a two-stage generation process, making it dependent on reconstruction quality and computationally expensive. In contrast, our method takes only a single satellite image as input and generates the entire sequence end-to-end. We introduce a triplane representation to encode the satellite scene and a Ray-Based Pixel Attention module

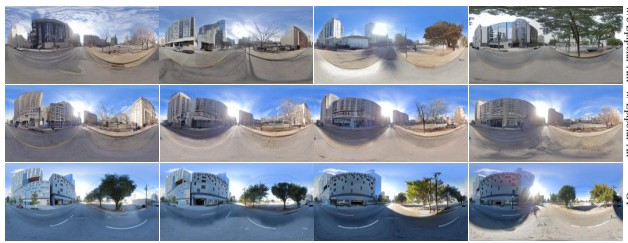

Figure 6: Qualitative comparison without (top) and with (middle) the proposed epipolar-constrained attention.

Table 4: Comparison of Full Cross-Attention and Epipolar-Constrained Temporal Attention for realism and temporal consistency.

|  | ↓FID | ↓DINO | ↓Depth | ↓FVD | ↓CLIPSIM |
|---|---|---|---|---|---|
| w/ Full Cross-Att | 42.60 | 4.253 | 6.231 | 2.150 | 7.516 |
| w/ Epipolar-Att | **27.41** | **4.156** | **5.623** | **2.101** | **6.820** |

Table 3: Application to the downstream cross-view localization task. Experimental evaluation on the VIGOR dataset reveals the average localization error before and after synthetic data training.

|  | ↓Aligned | ↓Unaligned |
|---|---|---|
| w/o synth data | 5.22 | 5.33 |
| w/ Ours | **4.99** | **5.11** |

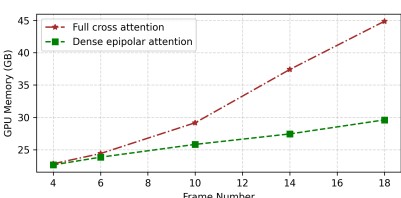

Figure 7: Memory comparison when generating different frame numbers in a video.

to explicitly enforce geometric consistency between the generated ground views and the satellite image. Furthermore, instead of using standard full cross-attention for multi-view consistency like Sat2GroundScape, we propose an epipolar-guided attention mechanism specifically designed for panoramic images, which greatly reduces computational cost (Figure 7) while achieving superior satellite-to-ground consistency and multi-view consistency (Table 4).

Moreover, their settings rely on multiple satellite images from different viewpoints or real ground-truth height maps, while ours requires only a single overhead satellite image as input.

Among these baselines, Sat2Density represents a Nerf-based method for cross-view generation. ControlS2S is a recent diffusion-based method that synthesizes ground-level images conditioned on a single satellite image. EscherNet is a state-of-the-art diffusion framework for multiview image generation. To adapt EscherNet to our satellite-to-ground task, we treat the satellite image as the reference view and assign it a virtual camera pose positioned above the image center and oriented vertically downward. Because satellite imagery is approximately orthographic, we model this virtual camera as being located at a high altitude to approximate its projection. Each ground-view camera pose in the trajectory is then expressed as a relative transformation with respect to this satellite reference pose. This formulation provides reference–target pose pairs in exactly the format required by EscherNet. For fair comparison, we retrain all methods on our proposed VIGOR++ dataset.

As shown in Table 1, EscherNet performs the worst across perceptual, semantic, pixel-wise, and depth-consistency metrics, mainly because it lacks an explicit mechanism to bridge the large domain gap between satellite and ground views. However, it achieves better multiview consistency (measured by FVD and CLIPSIM) than Sat2Density and ControlS2S, owing to its built-in multi-view coherence modeling. In contrast, SatDreamer360 explicitly addresses both cross-view appearance disparity and the challenge of multiview continuity. As a result, it achieves the best overall performance across all dimensions, combining high image fidelity with smooth and consistent video generation. Qualitative results in Figure 4 further support these findings. EscherNet, which relies on implicit scene encoding, struggles to produce realistic ground-level images. ControlS2S, as illustrated in Figure 1, lacks effective mechanisms for multiview consistency, leading to spatial discontinuities across frames. In comparison, SatDreamer360 preserves the underlying scene layout and produces ground-view sequences that are both spatially coherent and temporally smooth.

## 4.2 MODEL ANALYSIS

Our method consists of two key components: (1) a ray-guided cross-view feature conditioning mechanism that ensures geometric consistency between the satellite image and the generated ground views, and (2) an epipolar-constrained attention module that enforces multi-view consistency across frames in the generated ground-view sequences.

To validate the effectiveness of the proposed **Ray-Guided Cross-view Feature Conditioning Mechanism**, we conduct single-image satellite-to-ground generation experiments on CVUSA and VIGOR, removing the influence of sequential modeling and allowing direct comparison with state-of-the-art methods. We compare our method with Pix2Pix (Isola et al. (2017b)), S2S (Shi et al. (2022)), Sat2Density (Qian et al. (2023)), and ControlS2S (Ze et al. (2025)), while CrossDiff Li et al. (2024a) results are cited from the original paper. Note that pixel-wise metrics may not fully capture the quality of synthesized images in this task, as they are sensitive to factors such as lighting and sky appearance that are not explicitly modeled in satellite imagery. Quantitative and qualitative comparisons on both datasets (Table 2, Figure 5) clearly demonstrate that our method outperforms existing approaches in terms of overall generation quality and exhibits stronger left–right boundary consistency across the synthesized ground views. This performance gain primarily results from the proposed Ray-Based Pixel Attention module, which samples features along each viewing ray and explicitly incorporates geometric information from the satellite image. By doing so, the module ensures that the generated ground views are accurately aligned with the underlying satellite-view representation, preserving both local details and global spatial structure, and effectively mitigating artifacts that typically arise in boundary regions or complex geometries.

Next, we verify the necessity of the proposed **Epipolar-Constrained Temporal Attention**. As shown in Figure 6, incorporating this module significantly improves sequence consistency. Furthermore, Table 4 and Figure 7 demonstrate that replacing Epipolar-Constrained Attention with full cross-attention substantially increases computational cost, whereas our approach achieves both lower cost and stronger sequence consistency. The advantage stems from introducing geometric priors via epipolar geometry, which filters out irrelevant matches, suppresses noise propagation, and eliminates a large number of non-corresponding points before attention computation. Additionally, our sparse interframe attention, in which each frame attends only to its immediate neighbors, allows the model to scale efficiently to longer sequences without compromising performance.

**Application to Downstream Cross-View Localization Task.** SatDream360 can be leveraged to generate synthetic ground-view data from satellite imagery, enabling enhanced training for downstream tasks. We evaluate this benefit in the context of cross-view localization using the state-of-the-art G2SWeakly (Shi et al. (2024)) model as a baseline. To ensure fair comparison, we follow the same training configuration as the baseline: 10 epochs with identical batch sizes. The only modification is the inclusion of SatDream360-generated data for training augmentation. As shown in Table 3, the augmented model achieves superior performance, demonstrating that the high-fidelity, geometrically consistent samples produced by SatDream360 provide meaningful improvements for cross-view localization tasks.

## 5 CONCLUSION

We propose a novel framework for satellite-to-ground multiview generation, addressing the challenging task of synthesizing continuous ground-level panoramas from a single top-down satellite image. Our approach tackles both spatial and multiview consistency through two key modules: (1) a Ray-Guided Cross-View Feature Conditioning mechanism for accurately constructing satellite-and-ground-view correspondences, and (2) a Multi-scale Epipolar-Constrained Attention module that ensures multiview consistency with significantly reduced computational cost compared to standard cross-attention. To facilitate evaluation, we introduce VIGOR++, a large-scale benchmark dataset of aligned panoramic sequences and satellite views. Extensive experiments across multiple metrics and datasets demonstrate that our method outperforms state-of-the-art baselines in perceptual realism, semantic consistency, and multiview stability. We believe that this work provides a strong foundation for future research in cross-view generative modeling, with broad potential applications in 3D reconstruction, autonomous driving, and simulation environments.

### REPRODUCIBILITY STATEMENT

The implementation details of our model are provided in Section 3, with training settings and evaluation protocols provided in Section 4 and Appendix A.7. Additional ablation studies are included in the Appendix A.8, A.9, A.11 to clarify the effect of individual components. We promise to release both the dataset and the code to facilitate reproducibility.

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

# A   APPENDIX

## A.1   THE USE OF LARGE LANGUAGE MODELS (LLMS)

Large Language Models (LLMs) were employed solely as writing and editing assistants during manuscript preparation. In particular, we used an LLM to refine language, improve readability, and enhance clarity across various sections. Its contributions included tasks such as sentence rephrasing, grammar correction, and improving the overall coherence and flow of the text.

Importantly, the LLM played no role in the conception of research ideas, methodological design, or experimental execution. All core concepts, analyses, and results were entirely developed and validated by the authors. The LLM's involvement was strictly limited to linguistic refinement and did not influence the scientific content or data interpretation. The authors take full responsibility for the entirety of the manuscript, including any portions polished with LLM assistance. We have ensured that all LLM-generated content adheres to ethical standards and does not contribute to plagiarism or scientific misconduct.

## A.2   EXPLANATION OF PROJECTION GEOMETRY

We provide an explanation of Eq. 4, describing the correspondence between pixel coordinates in the camera coordinate system and the angles of the rays found. Any point $(u, v)$ in the pixel coordinate system corresponds to a camera ray with angles $(\psi, \theta)$, where $\psi$ is the yaw angle ranging from $[-\pi, \pi]$ and $\theta$ is the pitch angle ranging from $[-\pi/2, \pi/2]$. For example, as shown in Figure 8, the pixel at coordinates $(u', v')$ corresponds to the following ray angles:

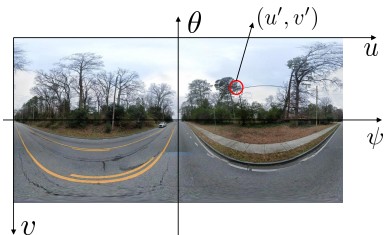

Figure 8: Correspondence between image pixel coordinates and camera ray angles.

$$\psi_{u',v'} = (u' - \frac{W}{2})/W \times 2\pi, \quad \theta_{u',v'} = (\frac{H}{2} - v')/H \times \pi, \tag{8}$$

where H and W are the height and width of the panoramic image.

## A.3   PROOF OF EQ. 6

**Prepare.** For vectors $a = [a_1\ a_2\ a_3]^T$ and $b = [b_1\ b_2\ b_3]^T$, the outer product of the two vectors is defined as:

$$a \otimes b = \begin{bmatrix} a_2b_3 - a_3b_2 \\ a_3b_1 - a_1b_3 \\ a_1b_2 - a_2b_1 \end{bmatrix} = \begin{bmatrix} 0 & -a_3 & a_2 \\ a_3 & 0 & -a_1 \\ -a_2 & a_1 & 0 \end{bmatrix} b = \hat{a}b. \tag{9}$$

Here, we introduce the skew-symmetric matrix notation, $\hat{a}$, which corresponds to the vector $a$. Using this notation, the outer product $a \otimes b$ is expressed compactly as the matrix-vector product $\hat{a}b$.

**Proof.** Consider the ground images of frames $m$ and $n$, there is a spatial point $Q = (x, y, z)$ in the coordinate system of frame $m$ corresponds to pixel points $g_{u,v}^m$ on frame $m$ and $g_{u',v'}^n$ on frame $n$ in panoramic images: These projections are given by:

$$g_{u,v}^m = P(Q), \quad g_{u',v'}^n = P(R_{mn}Q + t_{mn}), \tag{10}$$

where $P$ denotes the equirectangular camera projection transformation, $R_{mn}$ and $t_{mn}$ denote the relative rotation and translation between frames $m$ and $n$. Perform the inverse projection transformation to obtain:

$$P^{-1}(g_{u,v}^m) = Q, \quad P^{-1}(g_{u',v'}^n) = R_{mn}Q + t_{mn}. \tag{11}$$

Substitute $Q$ from the left equation into the right gives:

$$P^{-1}(g_{u',v'}^n) = R_{mn}(P^{-1}(g_{u,v}^m)) + t_{mn}. \tag{12}$$

By simultaneously left multiplying with skew-symmetric matrix $\hat{t}mn$ (corresponding to $t_{mn}$), as introduced in Eq. 9:

$$\hat{t}_{mn}(P^{-1}(g_{u',v'}^n)) = \hat{t}_{mn}R_{mn}(P^{-1}(g_{u,v}^m)). \tag{13}$$

Next, we multiply both sides on the left by the transpose $(p^{-1}(g_{u',v'}^n))^T$:

$$(P^{-1}(g_{u',v'}^n))^T \hat{t}_{mn}(P^{-1}(g_{u',v'}^n)) = (P^{-1}(g_{u',v'}^n))^T \hat{t}_{mn}R_{mn}(P^{-1}(g_{u,v}^m)). \quad (14)$$

Since the product$(P^{-1}(g_{u',v'}^n))^T \hat{t}_{mn}(P^{-1}(g_{u',v'}^n)) = 0$ (because $\hat{t}_{mn}(P^{-1}(g_{u',v'}^n))$ is orthogonal to $P^{-1}(g_{u',v'}^n)$), the above simplifies to:

$$(P^{-1}(g_{u',v'}^n))^T \hat{t}_{mn}R_{mn}(P^{-1}(g_{u,v}^m)) = 0. \quad (15)$$

This result implies that pixels $g_{u,v}^m$ on frame $m$ and $g_{u',v'}^n$ on frame $n$ that correspond to the same spatial point must satisfy this constraint relationship in Eq. 6. Therefore, during temporal attention, the points $g_{u,v}^m$ on frame $m$ only need to focus on the set of points $\{g_{u',v'}^n\}$ on frame $n$ that satisfy the above constraint. This significantly reduces the computational complexity compared to focusing on all pixels in the image.

### A.4 EPIPOLAR-CONSTRAINED IN PANORAMIC IMAGES

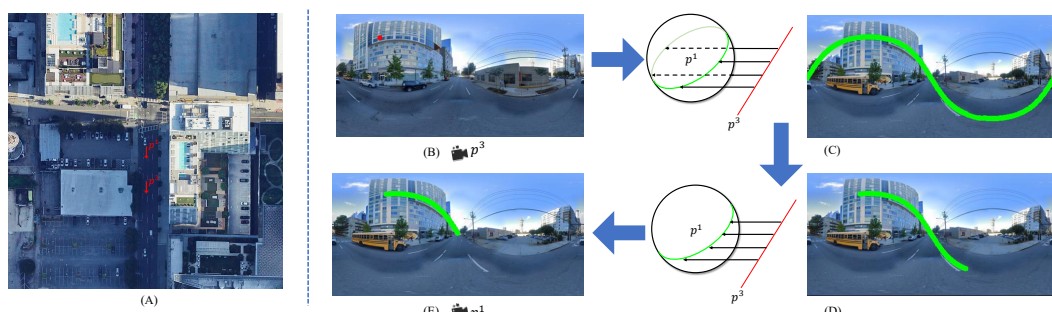

Figure 9: (A) Example of camera positions $p_3$ and $p_1$. (B) Selected point on source image plane $p_3$. (C) Epipolar ray projected onto target image plane $p_1$. (D) Simplified epipolar curve after removing non-intersecting portions. (E) potential correspondences refined by depth constraints.

In Figure 9, we show the full derivation of the panoramic epipolar geometry. Taking the camera positions $p^3$ and $p^1$ shown in Figure 9(A) as an example, we consider the red point on the image plane of $p^3$ in Figure 9(B). As illustrated in Figure 9(C), because $p^1$ uses an equirectangular projection, the ray corresponding to this point on $p^3$ is projected onto the image plane of $p^1$ as a green wavy curve. Since the back side of the sphere does not intersect with this ray, it can be omitted, resulting in the simplified curve shown in Figure 9(D). Moreover, the valid portion of the ray from $p^3$ must lie in front of the camera, so points corresponding to negative ray directions can be removed. This yields the final green line segment in Figure 9(E), which represents the valid epipolar region. The true correspondence of the red point on $p^3$ must lie on this green segment.

By using this formulation, we no longer need to compare every point on the image plane of $p^3$ with all points on $p^1$. Instead, attention is restricted to the green line segment in Figure 9(E). This significantly reduces computational complexity and avoids introducing noise through interactions with irrelevant points.

### A.5 SCALING TO LARGE-SCALE SCENES

As described in Section 3, the proposed model leverages Ray-Guided Cross-View Feature Conditioning and Epipolar-Constrained Attention to generate ground-view images aligned with the satellite inputs and consistent across views. However, scaling the approach to larger scenes remains challenging. Due to the inherent stochasticity of diffusion models, different batches conditioned on the same satellite image can produce noticeably different results, as illustrated in the first two rows of Figure 10. This highlights the need to establish connections across images generated from multiple batches to enhance the scalability of scene generation.

To incorporate information from previously generated images into the upcoming sequence, we update the triplane representation using Image Cross-Attention (ICA Li et al. (2024c)), which enriches each point on the XY, XZ, and YZ planes by referencing corresponding pixels from previously generated

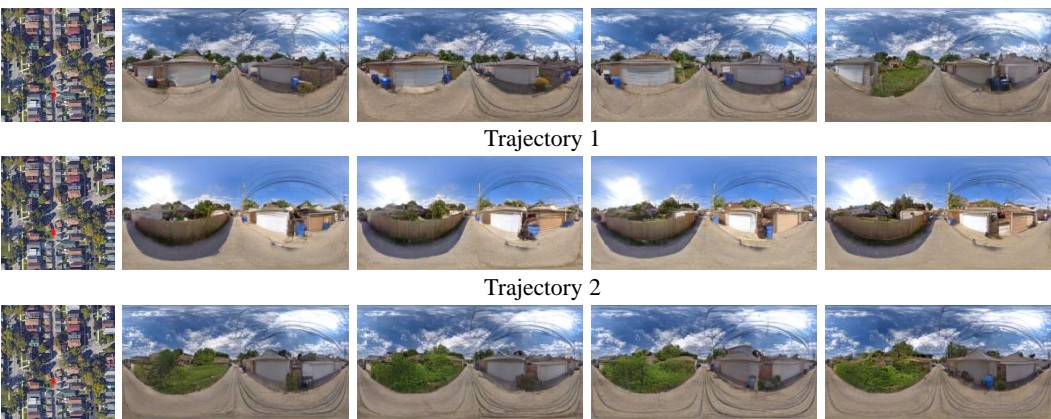

Trajectory 1

Trajectory 2

Trajectory 2 + Update the triplane with images generated in Trajectory 1

Figure 10: Using the same satellite image as a condition, different trajectories are input to generate corresponding images. From top to bottom: results for trajectory 1, results for trajectory 2, and results for trajectory 2 after updating the triplane with images generated from trajectory 1. Updating the triplane ensures that newly generated results are related to prior sequences.

images. For example, points on the XY plane are sampled along the $Z$-axis, projected into camera space, and aligned with image coordinates, enabling feature transfer through ICA:

$$F_{xy}^{\text{top}} = \text{ICA}\left(F_{xy}^{\text{top}}, \text{Ref}_{xy}^{2D}\right), \quad \text{Ref}_{xy}^{2D} = \{F_{u_i,v_i}^j\}. \tag{16}$$

Here, $(u_i, v_i) = P(R_j(x, y, z_i) + t_j)$ projects sampled 3D points into the $j$-th image plane via rotation $R_j$, translation $t_j$, and projection $P$, where $\{(x, y, z_i)\}$ represents the set of sampled points along the $Z$-axis. By integrating ICA with CVHA, the triplane accumulates information from prior results, yielding coherent and scalable scene generation across multiple batches (Figure 10, bottom row). By combining ICA and CVHA mechanisms, the triplane's features are enriched through the aggregation of information from previously generated images. As shown at the bottom of Figure 10, this approach enables the generation of extended image sequences across multiple batches, ensuring coherence and continuity in the outputs.

## A.6 DETAILS OF VIGOR++

The VIGOR++ dataset includes 91,498 satellite images and the same number of street-view images, as shown in Figure 11(a). These images are evenly distributed across ten cities, covering a total area of 117.47 km².

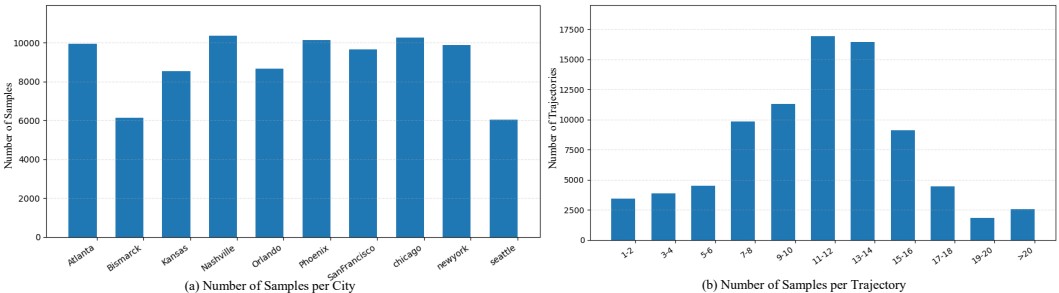

(a) Number of Samples per City

(b) Number of Samples per Trajectory

Figure 11: (a) Distribution of the Number of Samples Across Cities. (b) Distribution of Trajectory Sample Counts.

To construct trajectories, we first associate each ground-view image with its corresponding satellite patch, where the ground-view image lies at the center. Using this center point as the seed, we first filter out images captured under different weather conditions or on different dates, as these frames lack temporal consistency. We then construct a connectivity graph and apply depth-first search to extract all feasible routes. A semi-automatic procedure, combined with manual verification, then selects the longest and most temporally consistent route. Through this process, we obtain one trajectory for each satellite image.

Overall, we generate over 90,000 cross-view satellite–ground video pairs. Among them, 84,055 pairs are used for training and 7,443 pairs are used for testing. As shown in Figure 11(b), most trajectories consist of 7 to 16 ground-view frames. The average frame interval is approximately 11 m, with the smallest interval being 0.079 m and the largest reaching 20 m.

## A.7 ADDITIONAL EXPERIMENT DETAILS

Our model is fine-tuned based on the pre-trained Stable Diffusion 1.5 model (Rombach et al. (2022)), with the training process divided into three stages.

Initially, we train on single-image generation tasks with a batch size of 32 for 300 epochs, focusing on adapting the parameters of Ray-Guided Cross-View Feature Conditioning to generate ground-level images consistent with satellite geometry.

Next, we incorporate the Epipolar-Constrained Temporal Attention module to enable continuous ground scene generation This phase involves a total of 300 epochs. The first 5 epochs focus on pre-heating the Temporal Attention module with a batch size of 8 and 3 images per sequence. This is followed by fine-tuning the entire model for 200 epochs with the same batch size and sequence length, and then further fine-tuning for long sequence generation with a batch size of 4 and 5 images per sequence until reaching 300 epochs.

As the Autoencoder was originally trained only on single-image tasks, it caused flickering artifacts when handling a temporally coherent sequence of images. Same to (Blattmann et al. (2023b)), we maintain the original parameters and introduce temporal modules (3D convolution) in the decoder. The Autoencoder is trained on VIGOR++ with a batch size of 4 and 3 images per sequence for 40 epochs, focusing solely on training the added temporal module.

Throughout the training process, the learning rate is set to 7.0e-05, the optimizer used is AdamW, and all experiments are conducted on four NVIDIA L40 GPUs. This comprehensive training pipeline enables the model to generate geometrically accurate ground-level images and seamless, temporally coherent sequences.

## A.8 ABLATION STUDY ON SCENE REPRESENTATIONS

The comparison between triplane and BEV scene representations is presented in Table 5, where the BEV corresponds to using only the XY-plane from the triplane. The memory usage is measured with a batch size of 32.

Table 5: Quantitative comparison of Triplane and BEV(XY-plane) representations on the VIGOR dataset.

|  | $\downarrow P_{alex}$ | $\downarrow$DINO | $\downarrow$SegAny | $\uparrow$SSIM | $\uparrow$PSNR | $\uparrow$SD | $\downarrow$Depth | Memory |
|---|---|---|---|---|---|---|---|---|
| BEV | 0.3803 | 4.408 | 0.3549 | 0.4134 | 13.64 | 12.94 | 7.061 | **19011** |
| Triplane | **0.3469** | **4.287** | **0.3471** | **0.4385** | **14.08** | **13.11** | **6.727** | 20303 |

These results show that the triplane representation consistently outperforms the BEV (XY-plane) representation across all metrics, while introducing only a modest increase in computational overhead. The improvement comes from the triplane's ability to capture richer 3D structures. A pure XY-plane representation lacks vertical information, which is critical for rendering views under varying pitch angles. Moreover, our model samples features along camera rays. Relying solely on XY-plane features leads to incomplete spatial support, especially for oblique rays. The triplane effectively overcomes this limitation.

## A.9 ABLATION STUDY ON RAY-GUIDED CONDITION

To validate the impact of the ray-guided conditioning mechanism, we compare our method with a vanilla conditioning approach, which performs cross-attention between every 3D point feature in the triplane and each image pixel. The corresponding ablation results on VIGOR are shown in Figure 6:

The ray-guided conditioning substantially improves performance. This is because vanilla conditioning lacks explicit geometric constraints, often resulting in geometric distortions, whereas ray-guided conditioning leverages Ray-Based Pixel Attention to dynamically sample points along viewing rays

Table 6: Quantitative comparison of vanilla Cross-Attention and Ray-Based Pixel Attention on the VIGOR dataset.

| | $\downarrow P_{alex}$ | $\downarrow$DINO | $\downarrow$SegAny | $\uparrow$SSIM | $\uparrow$PSNR | $\uparrow$SD | $\downarrow$Depth | $\downarrow$Time(s) | $\downarrow$Memory(MB) |
|---|---|---|---|---|---|---|---|---|---|
| Vanilla Condition | 0.5413 | 5.425 | 0.3911 | 0.3174 | 12.35 | 12.05 | 25.38 | 120.28 | 22506 |
| Ray-guided condition | **0.3469** | **4.287** | **0.3471** | **0.4385** | **14.08** | **13.11** | **6.727** | **39.64** | **18296** |

using learnable offsets (Eq. 5). This approach ensures spatial coherence and improves geometric alignment across views.

Furthermore, we conduct an ablation study on the Dynamic Refinement of Offsets in Ray-Based Pixel Attention, with results shown in Table 7:

Table 7: Ablation study on the effect of dynamic refinement tested on the CVUSA dataset.

| | $\downarrow P_{alex}$ | $\downarrow$DINO | $\downarrow$SegAny | $\uparrow$SSIM | $\uparrow$PSNR | $\uparrow$SD | $\downarrow$Depth |
|---|---|---|---|---|---|---|---|
| w/o Dynamic | 0.4647 | 4.935 | 0.3639 | 0.3736 | 13.45 | 12.38 | 19.69 |
| w/ Dynamic | **0.4255** | **4.807** | **0.3602** | **0.3812** | **13.88** | **12.42** | **19.36** |

The goal of Ray-Based Pixel Attention is to aggregate meaningful features for each ground-level pixel by sampling along its corresponding 3D ray projected into the triplane. However, Early in the diffusion process, the latent features are dominated by noise, making accurate 3D correspondence difficult. To address this, we begin with uniform sampling along each ray. But uniform sampling often leads to sparse or suboptimal feature aggregation, leading to degraded performance, as shown in the first row of Table 7.

To mitigate this, we propose a mechanism to dynamically refine both the sampling offsets and their corresponding weights during diffusion inference. As denoising progresses, latent features gradually capture meaningful scene structure. The model uses these evolving features to predict offsets and weights for each sampling point along the ray, enabling more accurate and view-consistent feature aggregation. This leads to improved performance, as shown in the second row of Table 7.

## A.10 ABLATION STUDY ON EPIPOLAR-CONSTRAINED ATTENTION

To more thoroughly evaluate the contribution of the Epipolar-Constrained Attention mechanism to geometric consistency, we extend our ablation studies by removing this component and comparing it against a vanilla inter-frame attention baseline (Full Cross-Att).

Beyond FVD, we additionally measure the average inter-frame similarity using a CLIP-based metric (CLIPSIM). We also conduct a user study, where we compute the Average User Ranking (AUR). Specifically, we randomly sample 1,000 trajectories from the test set and collect rankings from five users to obtain the final AUR scores. The results are summarized below:

Table 8: Ablation Study of the Epipolar-Constrained Attention on the VIGOR++ Dataset.

| | $\downarrow$DINO | $\downarrow$SegAny | $\uparrow$PSNR | $\downarrow$Depth | $\downarrow$FVD | $\downarrow$CLIPSIM | $\uparrow$AUR(seq) |
|---|---|---|---|---|---|---|---|
| w/o Epipolar-Att | 4.2748 | 0.3626 | **12.86** | 6.174 | 3.439 | 10.20 | 0.174 |
| w/ Full Cross-Att | 4.253 | 0.3589 | 12.75 | 6.231 | 2.150 | 7.516 | 1.136 |
| w/ Epipolar-Att | **4.156** | **0.3563** | 12.75 | **5.623** | **2.101** | **6.820** | **1.690** |

These results demonstrate that adding Epipolar-Constrained Attention yields a substantial improvement over the variant without it. Compared with the vanilla inter-frame attention, our method achieves a 10% gain in CLIPSIM, receives significantly higher user rankings, and, as shown in Fig. 7, substantially reduces computational resources. Together, these findings provide strong evidence that the proposed Epipolar-Constrained Attention effectively enhances inter-frame geometric consistency.

## A.11 Ablation Study on sparse interframe attention

We adopt a sparse interframe strategy after careful comparison with the dense interframe strategy. Specifically, the sparse strategy queries only the two preceding frames for each target frame, whereas the dense strategy attends to all frames within the sequence. As shown in Table 9, both strategies achieve similar per-frame quality, but the sparse strategy yields better multiview consistency. This is because distant frames often contribute less meaningful information and may introduce noise, while nearby frames provide more relevant context.

Table 9: Ablation study on the effect of sparse interframe strategy tested on the VIGOR++ dataset.

|        | ↓FVD  | ↓CLIPSIM | ↓$P_{alex}$ | ↓DINO | ↓Depth | ↑PSNR |
|--------|-------|----------|-------------|-------|--------|-------|
| Dense  | 2.253 | 7.071    | 0.4085      | **4.153** | 5.791  | **12.91** |
| Sparse | **2.101** | **6.820** | **0.3955** | 4.156 | **5.623** | 12.75 |

We further compare runtime and memory usage as the number of frames increases (with batch size set to 1 and 50 DDIM steps). Table 10 shows that the sparse strategy is significantly more efficient, particularly for longer sequences.

Table 10: Ablation study on the resource consumption of the sparse interframe strategy.

| Strategy | Metric | 10 Frames | 20 Frames | 30 Frames |
|----------|--------|-----------|-----------|-----------|
| Dense    | Time(s) | 20.33 | 59.12 | 120.75 |
|          | Memory(MB) | 26174 | 31106 | 40520 |
| Sparse   | Time(s) | **14.29** | **22.50** | **32.71** |
|          | Memory(MB) | **25824** | **30242** | **35142** |

In summary, the sparse inter-frame strategy achieves better multiview consistency while reducing both computation time and memory usage, making it the preferred choice for our generation framework.

## A.12 Ablation study on satellite encoding models

We choose a CNN-based ResNet encoder because our task imposes strong spatial–geometric constraints, and the satellite features are expected to faithfully preserve the geometric structure of the overhead imagery. CNNs produce feature maps that are spatially continuous and locally coherent, making them naturally suitable for encoding geometry-aware local structures. In contrast, Vision Transformers partition the image into patches and apply global attention, a process that often disrupts the original geometric relationships in satellite imagery.

We also conduct an empirical comparison on the CVUSA dataset using ResNet and ViT for satellite encoding. As shown in the table 11, the ResNet encoder consistently outperforms ViT across all metrics:

Table 11: Ablation study on satellite encoding models.

|        | ↓P_squeeze | ↓P_alex | ↓DINO | ↓SegAny | ↑SSIM | ↑PSNR | ↑SD | ↓Depth |
|--------|------------|---------|-------|---------|-------|-------|-----|--------|
| Vit    | 0.3356 | 0.4469 | 5.049 | 0.3711 | 0.3371 | 13.30 | 12.11 | 20.96 |
| ResNet | **0.3146** | **0.4255** | **4.807** | **0.3602** | **0.3812** | **13.88** | **12.42** | **19.36** |

## A.13 influence of trajectory sequences

We analyze this effect of trajectory sequences from three perspectives: the trajectory length, the sampling interval, and the relative spatial relationship between the trajectory and the satellite image.

**Trajectory length.** We reorganize the test set by grouping trajectories based on their total length into four ranges: <20 m, 20–40 m, 40–60 m, and >60 m. The results are shown below. The model performs well for trajectories shorter than 60 m, but we observe noticeable degradation when the trajectory length exceeds 60 m. This limitation primarily arises from our inability to train the model

Table 12: Influence of trajectory length.

| Distance | ↓P_alex | ↓DINO | ↓SegAny | ↑SSIM | ↑PSNR | ↓Depth | ↓CLIPSIM |
|---|---|---|---|---|---|---|---|
| <20m | **0.3793** | 4.182 | **0.3515** | **0.4230** | **12.88** | 5.582 | 5.947 |
| 20-40m | 0.3812 | 4.159 | 0.3529 | 0.4141 | 12.79 | 5.422 | **5.921** |
| 40-60m | 0.3850 | **4.143** | 0.3538 | 0.4072 | 12.75 | **5.417** | 6.065 |
| >60m | 0.3972 | 4.160 | 0.3552 | 0.3839 | 12.69 | 5.760 | 6.744 |

on sufficiently long sequences due to computational constraints. We expect this issue to be mitigated by incorporating longer trajectories during training in future work.

**Sampling interval.** To examine how sampling density affects generation quality, we reorganize the test set and evaluate trajectories under different frame-interval ranges. Specifically, we group trajectories into two categories: intervals <10 m and intervals ≥10 m. The evaluation results are shown below. As the frame interval increases, the generation task becomes more challenging because the correlation between adjacent frames becomes significantly weaker. Consequently, we observe a consistent trend of performance degradation at larger sampling distances.

Table 13: Influence of sampling interval.

| Interval | ↓P_alex | ↓DINO | ↓SegAny | ↑SSIM | ↑PSNR | ↓Depth | ↓CLIPSIM |
|---|---|---|---|---|---|---|---|
| <10m | **0.3611** | **4.088** | **0.3420** | **0.4592** | **13.35** | **4.304** | **6.204** |
| ≥10m | 0.3848 | 4.155 | 0.3530 | 0.4090 | 12.82 | 5.494 | 6.639 |

**Relative spatial relationship between the trajectory and the satellite image.** To evaluate how the relative spatial relationship between the trajectory and the satellite image affects generation quality, we shift the satellite image so that the trajectory appears at different locations within it. For example, when the satellite image is translated 20 m to the right, the trajectory—originally centered—becomes offset by 20 m. Using this setup, we test offsets of ±20 m and ±40 m. The results show that the generation quality is almost unaffected by these spatial shifts. This is primarily because our method does not naively use the entire satellite image as a global conditioning input. Instead, we extract geometry-aware features through Ray-Based Pixel Attention, which selectively samples informative points from the satellite representation along the ray directions. This sampling-based design effectively mitigates sensitivity to the global spatial alignment between the trajectory and the satellite image.

Table 14: Influence of relative spatial relationship between the trajectory and the satellite image.

| Offset | ↓P_alex | ↓DINO | ↓SegAny | ↑SSIM | ↑PSNR | ↓Depth | ↓CLIPSIM |
|---|---|---|---|---|---|---|---|
| 0 | **0.3955** | 4.156 | 0.3563 | **0.3964** | 12.75 | **5.623** | 6.820 |
| ±20 | **0.3955** | **4.150** | **0.3560** | 0.3962 | 12.78 | 5.631 | 6.745 |
| ±40 | 0.3973 | 4.164 | 0.3567 | 0.3959 | 13.73 | 5.674 | **6.706** |

## A.14 HYPERPARAMETER ANALYSIS

**The number of ray samples(K).** We evaluate the effect of different numbers of ray samples K on the CVUSA dataset using a batch size of 16. The results are shown in Table 15. Although increasing the number of ray samples(K=12) allows the model to capture more features, the improvement over K=8 is relatively marginal. Considering that K=8 offers a significantly better balance between performance and memory consumption, we adopt K=8 as our default setting.

**The number of sampled points satisfying epipolar constraints(M).** We evaluate the influence of the number of epipolar sampling points on VIGOR++, and the results are summarized in Table 16. The experiments show that sampling four points already provides a strong geometric prior that effectively improves multi-view consistency. Increasing the number of samples offers only marginal gains while introducing substantial additional computational cost. Therefore, we adopt four sampling points as our default setting.

Table 15: Ablation study on the number of ray samples(K).

| K | ↓P_squeeze | ↓P_alex | ↓DINO | ↓SegAny | ↑SSIM | ↑PSNR |
|---|---|---|---|---|---|---|
| 4 | 0.3246 | 0.4386 | 4.920 | 0.3670 | 0.3606 | 13.62 |
| 8 | 0.3146 | 0.4255 | **4.807** | **0.3602** | 0.3812 | 13.88 |
| 12 | **0.3109** | **0.4206** | **4.807** | 0.3623 | **0.3708** | **13.91** |

Table 16: Ablation study on the number of sampled points satisfying epipolar constraints(M).

| M | ↓FVD | ↓CLIPSIM |
|---|---|---|
| 1 | 5.509 | 7.142 |
| 4 | **2.101** | 6.820 |
| 8 | 3.532 | **6.770** |

**Triplane resolution.** We extract features from 256×256 satellite images to a 32×32 resolution, maintaining an 8× downsampling rate consistent with the VAE encoder of the latent diffusion model. Preserving this feature resolution ensures that the triplane conditioning information is spatially aligned with the diffusion latent, which improves conditional guidance and training stability.

**Training design.** Empirically, the model reaches convergence after 300 epochs in each stage. In the first stage, it primarily learns consistency between ground views and the satellite image, while in the second stage, the Epipolar-Constrained Temporal Attention module is introduced to facilitate multi-view consistency. Convergence is assessed based on the clarity and structural coherence of generated samples, absence of obvious noise or artifacts, and stabilization of validation metrics such as SSIM and $P_{alex}$ stabilize. The learning rate is set to 7e-05, following the successful practice of ControlS2S, which ensures stable training and consistent convergence. Finally, the batch size is gradually reduced from $32 \rightarrow 8 \rightarrow 4$ across stages due to GPU memory constraints: as the number of frames per sequence increases from $1 \rightarrow 3 \rightarrow 5$, memory consumption rises, necessitating smaller batches to accommodate longer sequences.

### A.15 USER STUDY

To provide a more comprehensive evaluation of our method, we conduct a user study. Specifically, we compute the Average User Ranking (AUR). We randomly sample 1,000 trajectories from the test set and collect rankings from five users to obtain the final AUR scores. Each user ranks the methods for each sequence: the top-ranked method receives 2 points, the second receives 1 point, and the third receives 0 points. The final AUR is computed by averaging the scores across users. The evaluation considers two aspects: similarity between the generated ground scenes and satellite images (S2G-Sim) and multi-view consistency (MV-Cons). These results, shown in Table 17, demonstrate that our method generates ground scenes that better reflect the satellite imagery and achieves superior multi-view consistency.

Table 17: User study.

| | ↑S2G-Sim | ↑MV-Cons |
|---|---|---|
| EscherNet | 0.481 | 0.974 |
| ControlS2S | 1.057 | 0.051 |
| Ours | **1.462** | **1.975** |

### A.16 EVALUATION OF GENERATION RESULTS UNDER DIFFERENT RANDOM SEEDS

To avoid potential bias, we conduct additional tests under different random noise seeds. On the VIGOR++ dataset, we evaluate our method using seeds 1, 25, 50, and 75. The corresponding results are presented in Table 18.

Table 18: Evaluation of generation results under different random seeds.

| Seed | ↓P_alex | ↓FID | ↓DINO | ↓SegAny | ↑SSIM | ↑PSNR | ↓Depth | ↓FVD | ↓CLIPSIM |
|---|---|---|---|---|---|---|---|---|---|
| 1 | 0.3955 | 27.39 | 4.157 | 0.3565 | 0.3965 | 12.75 | 5.616 | 2.127 | 6.815 |
| 25 | 0.3955 | 27.41 | 4.156 | 0.3563 | 0.3964 | 12.75 | 5.623 | 2.101 | 6.820 |
| 50 | 0.3954 | 27.52 | 4.155 | 0.3562 | 0.3966 | 12.76 | 5.620 | 2.126 | 6.798 |
| 75 | 0.3958 | 27.41 | 4.157 | 0.3564 | 0.3964 | 12.76 | 5.637 | 2.109 | 6.843 |
| Average | 0.3956 | 27.43 | 4.156 | 0.3564 | 0.3965 | 12.76 | 5.624 | 2.116 | 6.819 |
| Std | 0.0002 | 0.0591 | 0.0010 | 0.0001 | 0.0001 | 0.0058 | 0.0091 | 0.0128 | 0.0186 |

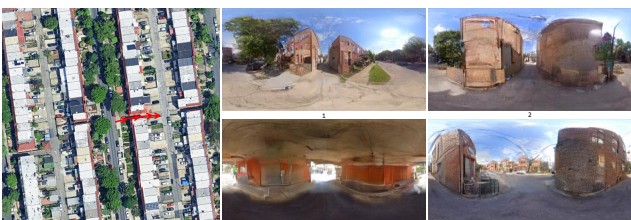

Figure 12: Failure case.

## A.17 LIMITATIONS AND DISCUSSIONS

While SatDreamer360 can generate continuous ground scene images from a single satellite image and a given ground camera trajectory, it still faces several limitations and corresponding areas for improvement.

Although VIGOR++ covers diverse regions, it is still constrained by Google Maps coverage and existing road networks, which may limit generalization to off-road or unstructured environments. As shown in Figure 12, our method occasionally produces incorrect results when generating scenes in narrow alleyways that are not accessible to vehicles. Future work will incorporate additional data sources such as drone imagery, vehicle-mounted cameras, or crowd-sourced panoramic data to expand coverage and enhance the model's generalization ability.

Additionally, Our method focuses primarily on modeling the static components of the scene and does not explicitly handle dynamic objects such as vehicles and pedestrians. As a result, the generated sequences may lack realistic dynamic behaviors. In future work, we plan to incorporate dynamic objects to establish more realistic satellite-to-ground correspondences.

## A.18 BROADER SOCIAL IMPACTS

Our proposed system, SatDreamer360, can generate continuous ground scene images from a single satellite image and a given ground camera trajectory, making it a valuable tool for applications such as 3D reconstruction, simulation, and autonomous navigation. However, while it can produce visually plausible ground scenes, it still struggles to capture all real-world details, and caution should be exercised when deploying it in safety-critical scenarios.

Moreover, like many generative models, SatDreamer360 could be misused to synthesize misleading or fake visual content. To mitigate such risks, we recommend using it only in controlled research or industrial settings, incorporating usage licenses and watermarking techniques to trace generated content, and clearly disclosing when images are synthetic. These safeguards can help prevent misuse and ensure that the technology is applied responsibly.

## A.19 EXAMPLES OF SATELLITE IMAGES AND TRAJECTORIES

Figure 13 presents example satellite images along with their corresponding trajectories, corresponding to the figure 4 shown in the main text.

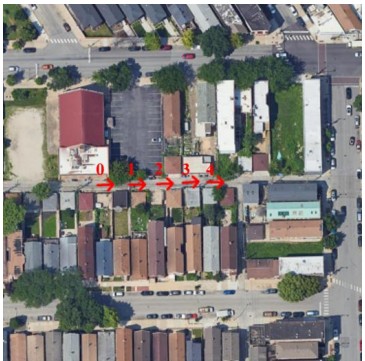 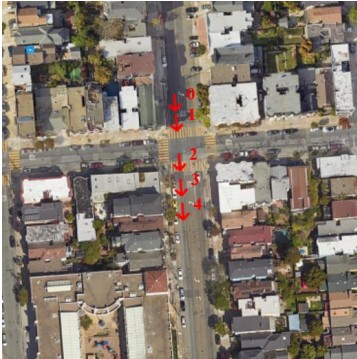

Trajectory 1 (T1)                    Trajectory 2 (T2)

Figure 13: Satellite images and corresponding trajectory points.

## A.20 MORE VISUALIZATION RESULTS

In Figure 14, we provide additional visualization results. Our method accurately follows the geometric layout of the satellite imagery while maintaining strong multiview consistency. It achieves robust performance both in densely built urban areas and rural regions. Even in challenging turning scenarios (middle of the figure), the generated results exhibit good continuity.

To further evaluate the generalization capability of our method, we test our model on cities in Africa, and Europe, despite training exclusively on U.S. cities. Under this challenging cross-continent setting, as shown in Figure 15 and Figure 16, other methods often fail, whereas our approach consistently produces plausible and coherent results, demonstrating strong generalization ability. This robustness stems from the tri-plane representation, which captures both horizontal and vertical structural cues across multiple orthogonal planes, providing sufficient geometric support for a wide variety of urban environments. Moreover, our Ray-Based Pixel Attention effectively aligns the ground-view generation with the satellite representation, enabling the model to maintain geometric consistency with the satellite imagery across diverse city layouts.

To evaluate the model's generation capability on long trajectories, we provide additional visualizations in Figure 17. The images are arranged from left to right and top to bottom, showing the generated results of a vehicle navigating a left-turn scenario. The entire trajectory spans approximately 70 meters, and the results demonstrate that our method maintains high geometric consistency even for longer and curved paths.

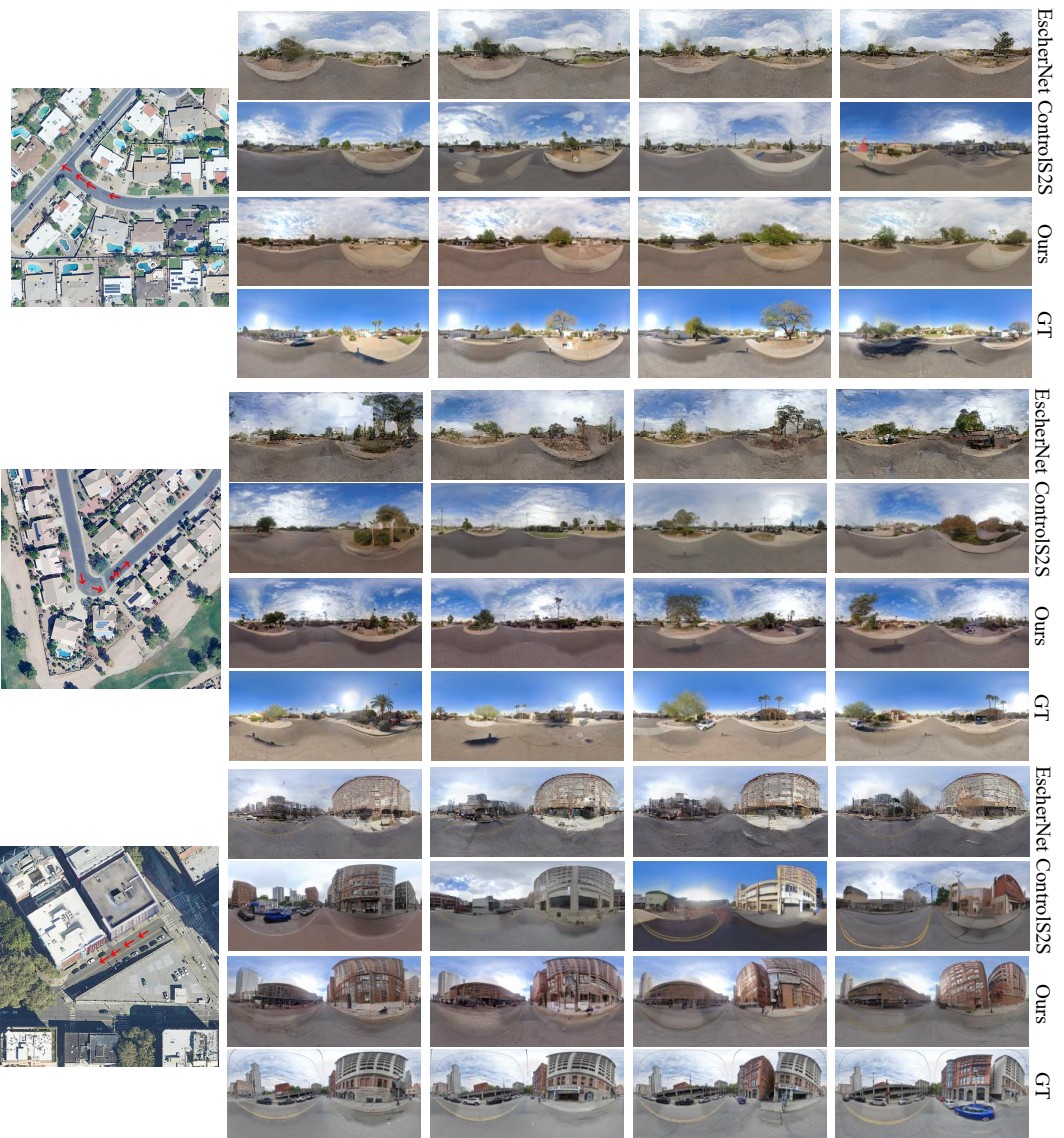

Figure 14: Qualitative comparison of ground-level image sequences along trajectories. The left shows the satellite image and the corresponding trajectory, while the ground-level images progress along the trajectory from left to right.

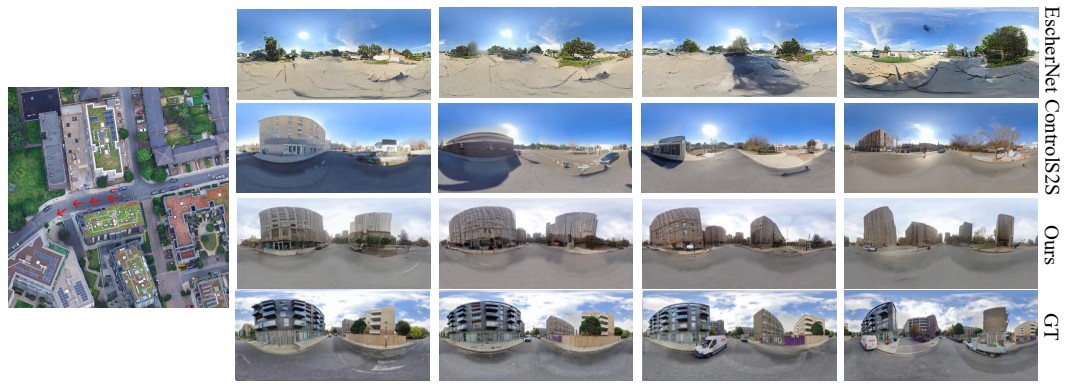

Figure 15: Results of ground-view generation using European satellite images as input.

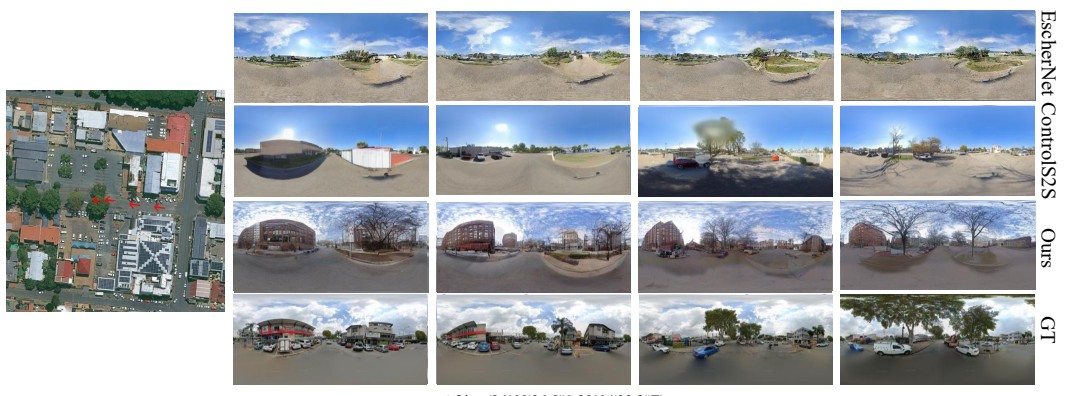

Figure 16: Results of ground-view generation using African satellite images as input.

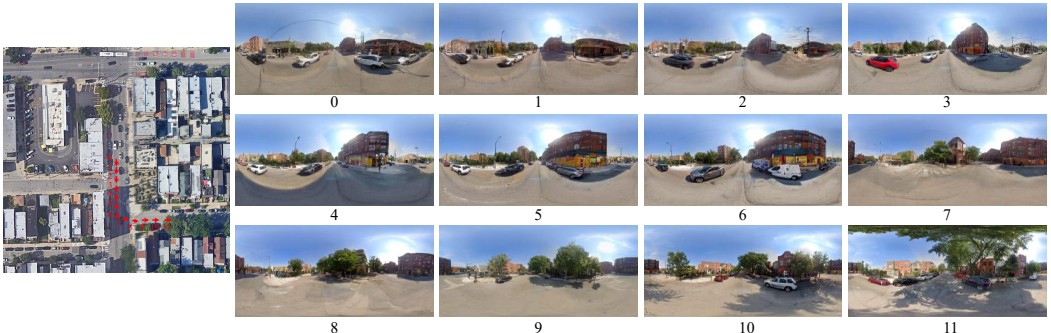

Figure 17: Generation results on long trajectories, where the images are ordered from left to right and top to bottom following the vehicle's forward motion.

