# OpenReview forum: "SatDreamer360: Multiview-Consistent Generation of Ground-Level Scenes from Satellite Imagery"
_ICLR.cc/2026/Conference — ICLR 2026 Poster_

### Official Review · Reviewer_5Zje · 2025-10-26

**Soundness:** 4
**Presentation:** 4
**Contribution:** 3
**Rating:** 8
**Confidence:** 3

**Summary:**

This paper proposes a framework for generating multiview-consistent 360° ground-level panoramic sequences from a single satellite image and a predefined camera trajectory. The two key methodological contributions are: 1. ray-guided cross-view feature conditioning (a ray-based pixel attention mechanism samples features along viewing rays, establishing explicit geometric correspondences between satellite and ground views), 2. epipolar-constrained attention for panoramas (an attention mechanism that enforces multiview consistency across frames by restricting attention to geometrically valid correspondences, reducing computational complexity). The paper also introduces the VIGOR++ Dataset. Experiments demonstrate state-of-the-art performance across perceptual, semantic, pixel-level, and multiview consistency metrics, outperforming methods like Sat2Density, ControlS2S, and EscherNet.

**Strengths:**

Novel Problem Formulation: While satellite-to-ground synthesis exists, the paper uniquely formulates the problem as generating geometrically consistent multiview sequences from a single satellite image with trajectory control. This is more challenging and practical than prior single-view or multi-satellite approaches. Most work targets perspective images; the consistent treatment of equirectangular projections throughout the pipeline is less explored.

Ray-based pixel attention with learnable offsets: Starting with uniform sampling and progressively learning offsets and weights as denoising proceeds is an elegant solution to the "noisy latent" problem in early diffusion steps. This bridges geometric sampling with diffusion's iterative nature.

Extending epipolar geometry from pinhole to equirectangular projections and integrating into attention mechanism requires non-trivial mathematical reformulation.

Triplane + diffusion integration: While triplanes exist, conditioning latent diffusion with ray-sampled triplane features (rather than global or BEV features) is novel. The cross-view hybrid attention for enriching planes with orthogonal projections adds spatial reasoning.

VIGOR++ represents substantial effort: 90K+ pairs, 10 cities, semi-automatic trajectory construction with manual refinement. The methodology for building continuous sequences from Street View (connectivity graphs, DFS path extraction) shows engineering creativity.

**Weaknesses:**

It is unfortunate that Deng et al. (2024) and Xu & Qin (2025) do not provide source code yet. They both appear quite strongly relevant so further discussion of input requirements (single vs. multi-view satellite), architectural differences, computational costs, and theoretical advantages would improve the paper.

Methods like Stable Video Diffusion or AnimateDiff could be adapted by conditioning on satellite images. These would test whether the geometric reasoning actually provides value over general temporal coherence mechanisms.

EscherNet seems like a weak baseline comparison. It is designed for object-centric multiview synthesis, not cross-view scene generation. The paper states they "adapt it to our setting by treating the satellite image as the source view" but provides no implementation details. This adaptation may be fundamentally disadvantaging EscherNet.

All metrics reported as single numbers without error bars, confidence intervals, or variance. Diffusion models are stochastic - different random seeds produce different results. So it would be good to have some sense of whether results are statistically significantly better.

There are quite a few hyperparameters without strongly justified values:
K (number of ray samples): value never specified in main paper
M (epipolar constraint points): "M ≪ HW" but exact value unclear
Why 2 preceding frames in sparse attention? No ablation on alternatives
Triplane resolution not specified until limitations section
Why 300 epochs for each stage? Was convergence analyzed?
Learning rate 7e-05 - was this tuned or borrowed from SD1.5?
Why batch size 32 → 8 → 4 across stages?

**Questions:**

Based on my comments above, questions that the authors could respond to in the rebuttal:

1. Statistical significance of results
2. Deeper methodological comparitive discussion to Deng et al. (2024) and Xu & Qin (2025)
3. Justification of hyperparameter choices
4. Sparse interframe attention - why 2 frames?
5. Provide additional EscherNet adaptation details
6. Why ResNet for satellite encoding? Have you tested modern alternatives like Vision Transformer?

---

> ### Author Response · Authors · 2025-11-22
> **Response to Reviewer 5Zje (Part 1/3)**
>
> We thank Reviewer-5Zje for the valuable comments. Please find our response below.
>
> > **Q1. Statistical significance of results.**
>
> We thank the reviewer for the suggestion. To avoid potential bias, we conduct additional tests under different random noise seeds. On the VIGOR++ dataset, we evaluate our method using seeds 1, 25, 50, and 75. The results are summarized in Tab. 18 of App. A.16 and are also shown below:
>
> | Seed | ↓P_alex | ↓FID  | ↓DINO | ↓SegAny | ↑SSIM  | ↑PSNR | ↓Depth | ↓FVD  | ↓CLIPSIM |
> |------|---------|-------|-------|---------|--------|-------|--------|-------|-----------|
> | 1    | 0.3955  | 27.39 | 4.157 | 0.3565  | 0.3965 | 12.75 | 5.616  | 2.127 | 6.815     |
> | 25   | 0.3955  | 27.41 | 4.156 | 0.3563  | 0.3964 | 12.75 | 5.623  | 2.101 | 6.820     |
> | 50   | 0.3954  | 27.52 | 4.155 | 0.3562  | 0.3966 | 12.76 | 5.620  | 2.126 | 6.798     |
> | 75   | 0.3958  | 27.41 | 4.157 | 0.3564  | 0.3964 | 12.76 | 5.637  | 2.109 | 6.843     |
> | Average | 0.3956  | 27.43 | 4.156 | 0.3564  | 0.3965 | 12.76 | 5.624  | 2.116 | 6.819     |
> | Std     | 0.0002  | 0.0591 | 0.0010 | 0.0001 | 0.0001 | 0.0058 | 0.0091 | 0.0128 | 0.0186 |
>
>
> > **Q2. Deeper methodological comparitive discussion to Deng et al. (2024) and Xu & Qin (2025.).**
>
> StreetScape (Deng et al., 2024) simplifies the task by providing a GT height map. With explicit geometric information available, the model no longer needs to infer the underlying 3D structure, effectively reducing the problem to one of surface-appearance synthesis. However, its reliance on accurate height maps is a critical limitation: such maps are difficult to obtain in practice, and any inaccuracy fundamentally breaks the pipeline. In addition, to maintain frame-to-frame continuity, StreetScape warps the previous frame using the height map and uses the warped result as a prior for generating the next frame. However, this design implies that generating each new frame requires an additional inference process, leading to high computational cost and the accumulation of temporal errors over long sequences.
>
> Similarly, Sat2GroundScape (Xu & Qin, 2025) also depends on additional conditions as input. The method first reconstructs a regional 3D model from multiple satellite images captured at different viewpoints using a stereo-matching approach, and uses only the rendered ground views from the 3D model as conditions. While this design reduces the problem to a super-resolution task, it also makes the method heavily dependent on the quality of the reconstructed 3D model. To enforce multi-view consistency, it adopts a two-stage generation strategy: it first generates the initial frame and then uses full cross-attention to transfers its style to all rendered ground views. This pipeline introduces additional computational overhead and remains ultimately constrained by the quality of the reconstructed 3D model.
>
> In contrast, our method takes only a single satellite image as input and generates the entire sequence end-to-end. We introduce a triplane representation to encode the satellite scene and a Ray-Based Pixel Attention module to explicitly enforce geometric consistency between the generated ground views and the satellite image. Furthermore, instead of using standard full cross-attention for multi-view consistency like Sat2GroundScape, we propose an epipolar-guided attention mechanism specifically designed for panoramic images, which greatly reduces computational cost (Fig. 7) while achieving superior satellite-to-ground consistency and multi-view consistency (Tab. 4).

---

> ### Author Response · Authors · 2025-11-22
> **Response to Reviewer 5Zje (Part 2/3)**
>
> > **Q3. Justification of hyperparameter choices.**
>
> Thank you for pointing this out. We conduct additional experiments to analyze how hyperparameter influences generation quality, and we include these analyses in App. A.14.
>
> **(Q3.a) The number of ray samples(K)**\
> We evaluate the effect of different numbers of ray samples K on the CVUSA dataset using a batch size of 16. The results are shown below. Although increasing the number of ray samples(K=12) allows the model to capture more features, the improvement over K=8 is relatively marginal. Considering that K=8 offers a significantly better balance between performance and memory consumption, we adopt K=8 as our default setting.
>
> | K   | ↓$P_{squeeze}$ | ↓$P_{alex}$ | ↓DINO | ↓SegAny | ↑SSIM  | ↑PSNR |
> |-----|------------|---------|-------|---------|--------|-------|
> | 4   | 0.3246     | 0.4386  | 4.920 | 0.3670  | 0.3606 | 13.62 |   |
> | 8   | 0.3146     | 0.4255  | **4.807** | **0.3602**  | **0.3812** | 13.88 |
> | 12  | **0.3109**     | **0.4206**  | **4.807** | 0.3623  | 0.3708 | **13.91** |
>
> **(Q3.b) The number of sampled points satisfying epipolar constraints(M)**\
> We evaluate the influence of the number of epipolar sampling points on VIGOR++, and the results are summarized below. The experiments show that sampling four points already provides a strong geometric prior that effectively improves multi-view consistency. Increasing the number of samples offers only marginal gains while introducing substantial additional computational cost. Therefore, we adopt four sampling points as our default setting.
> | M | ↓FVD | ↓CLIPSIM |
> |---|------|-----------|
> | 1 | 5.509 | 7.142     |
> | 4 | **2.101** | 6.820     |
> | 8 | 3.532 | **6.770**     |
>
> **\(Q3.c) Triplane resolution**\
> We extract features from 256×256 satellite images to a 32×32 resolution, maintaining an 8× downsampling rate consistent with the VAE encoder of the latent diffusion model. Preserving this feature resolution ensures that the triplane conditioning information is spatially aligned with the diffusion latent, which improves conditional guidance and training stability. Due to time and resource constraints, we are currently unable to provide additional ablation experiments with other resolutions, but we plan to include such experiments in the final version.
>
> **(Q3.d) Why 300 epochs for each stage? Was convergence analyzed?**\
> Yes, empirically, the model reaches convergence after 300 epochs in each stage. In the first stage, the model primarily learns the consistency between ground views and the satellite image. In the second stage, we introduce the Epipolar-Constrained Attention module to facilitate multi-view consistency learning. We evaluate convergence separately for each stage. The convergence criteria include whether the generated samples are clear, structurally coherent, and free of obvious noise or artifacts, as well as whether metrics on a validation subset (a small portion of the training set), such as SSIM and $P_{alex}$ stabilize.
>
> **(Q3.e) Why learning rate is 7e-05?**\
> We set the learning rate to 7e-5 following the successful practice of ControlS2S. This learning rate ensures stable training and consistent convergence.
>
> **(Q3.f) Why batch size 32 → 8 → 4 across stages?**\
> This is primarily due to GPU memory limitations. As we gradually increase the number of frames in each sequence from 1 → 3 → 5, the memory consumption correspondingly rises. To accommodate these longer sequences during training, we reduce the batch size at each stage.
>
> > **Q4. Sparse interframe attention - why 2 frames?**
>
> We select two frames because one is a distant frame (the very first frame) and the other is an adjacent frame (the previous frame). The distant frame helps maintain global consistency across the sequence, focusing on preserving large-scale features such as the sky, while the adjacent frame enforces local consistency, ensuring coherence between consecutive views. Unlike typical video generation tasks, our task generates each frame based on camera pose. When the frame interval is large, attending to a long sequence is not meaningful, as the two views may have little to no overlap. Performing cross-attention with distant, non-overlapping frames does not help maintain multiview consistency. We conduct an experiment comparing attention to all frames versus attention to only two frames:
>
> | Frames | FVD↓   | CLIPSIM↓ |
> |---------|--------|----------|
> | all     | 2.253  | 7.070    |
> | 2       | **2.101**  | **6.820**    |
>
> The results show that focusing on only two frames yields better performance.

---

> ### Author Response · Authors · 2025-11-22
> **Response to Reviewer 5Zje (Part 3/3)**
>
> > **Q5. Provide additional EscherNet adaptation details.**
>
> EscherNet primarily controls the relative viewing geometry between generated views by conditioning on reference images and their corresponding camera poses. Its key idea is to embed the camera rotation and translation matrices into a transformer-based architecture, enabling the model to synthesize a target view at a specified pose.
>
> To adapt EscherNet to our satellite-to-ground task, we treat the satellite image as the reference view and assign it a virtual camera pose positioned above the image center and oriented vertically downward. Because satellite imagery is approximately orthographic, we model this virtual camera as being located at a high altitude (e.g., 500 m) to approximate its projection. Each ground-view camera pose in the trajectory is then expressed as a relative transformation with respect to this satellite reference pose. This formulation provides reference–target pose pairs in exactly the format required by EscherNet.
>
> During training, we feed EscherNet the satellite reference image together with its pose, and specify the target ground-view pose as the desired output pose. The corresponding real ground-view image is used as supervision to train the model. This adaptation ensures that EscherNet follows the same pose-conditioning protocol as our method, enabling a fair, controlled, and meaningful comparison under identical input settings.
>
>
> > **Q6. Why ResNet for satellite encoding? Have you tested modern alternatives like Vision Transformer?**
>
> We choose a CNN-based ResNet encoder because our task imposes strong spatial–geometric constraints, and the satellite features are expected to faithfully preserve the geometric structure of the overhead imagery. CNNs produce feature maps that are spatially continuous and locally coherent, making them naturally suitable for encoding geometry-aware local structures. In contrast, Vision Transformers partition the image into patches and apply global attention, a process that often disrupts the original geometric relationships in satellite imagery.
>
> We also conduct an empirical comparison on the CVUSA dataset using ResNet and ViT for satellite encoding. As shown in the table below, the ResNet encoder consistently outperforms ViT across all metrics:
>
> | Method | ↓$P_{squeeze}$ | ↓$P_{alex}$ | ↓DINO | ↓SegAny | ↑SSIM  | ↑PSNR | ↑SD    | ↓Depth |
> |--------|------------|---------|-------|---------|--------|-------|--------|--------|
> | Vit    | 0.3356     | 0.4469  | 5.049 | 0.3711  | 0.3371 | 13.30 | 12.11  | 20.96  |
> | ResNet | **0.3146**     | **0.4255**  | **4.807** | **0.3602**  | **0.3812** | **13.88** | **12.42**  | **19.36**  |
>
> > **Q7. Methods like Stable Video Diffusion or AnimateDiff could be adapted by conditioning on satellite images. These would test whether the geometric reasoning actually provides value over general temporal coherence mechanisms.**
>
> We thank the reviewer for the suggestion. Stable Video Diffusion (SVD) and AnimateDiff are both highly successful video diffusion models. Due to time and resource constraints, we currently provide additional experiments only for SVD. We extract satellite image features using CLIP and control ground-view generation via cross-attention. For ground-view camera poses, we follow the approach of MVDream[A], adding camera embeddings to time embeddings as residuals, and train the full model end-to-end.
>
> So far, we train the model for 50 epochs, and the results are as follows:
>
> | Method | ↓$P_{alex}$ | ↓FID  | ↓DINO | ↓SegAny | ↑SSIM  | ↑PSNR | ↓Depth | ↓FVD  | ↓CLIPSIM |
> |--------|---------|-------|-------|---------|--------|-------|--------|-------|-----------|
> | SVD    | 0.4698  | 46.45 | 4.679 | 0.3795  | 0.3651 | 11.78 | 6.809  | 5.487 | 7.288     |
> | Ours   | **0.3955**  | **27.41** | **4.156** | **0.3563**  | **0.3964** | **12.75** | **5.623**  | **2.101** | **6.820**     |
>
> Currently, SVD has not fully converged. We plan to continue these experiments and will include the updated results in the final version.
>
> [A] Mvdream: Multi-view diffusion for 3d generation

---

### Official Review · Reviewer_QDgj · 2025-10-30

**Soundness:** 3
**Presentation:** 3
**Contribution:** 3
**Rating:** 4
**Confidence:** 4

**Summary:**

This paper introduces SatDreamer360, a diffusion-based framework that generates multiview-consistent ground-level panoramas conditioned on a single satellite image.
The model integrates three main components:
Ray-guided cross-view feature conditioning, leveraging a triplane representation to model geometric correspondences between satellite and ground perspectives.
Epipolar-constrained attention, enforcing temporal and geometric consistency across panoramic frames using equirectangular projection geometry.
A new dataset, VIGOR++, extending the VIGOR dataset with continuous ground-level sequences and trajectory annotations for evaluating satellite-to-ground synthesis.

**Strengths:**

1.	Satellite-to-ground image generation is a challenging and underexplored topic. It has practical value for simulation, urban modeling, and cross-view localization.
2.	The combination of triplane representation, ray-guided sampling, and epipolar attention is technically sound. The pipeline effectively bridges satellite conditioning and panoramic generation.
3.	The results show noticeable improvements in both image realism and geometric coherence. The qualitative examples look visually convincing.

**Weaknesses:**

1.	The core architectural ideas—triplane representation (EG3D), ray-based sampling (MVDream, Zero123++), and epipolar-constrained attention (EpiDiff)—are largely from prior work. SatDreamer360 primarily integrates these components within a diffusion framework for a new application. While the system integration is well-executed, it lacks a fundamentally new algorithmic or theoretical contribution.
2.	Ablation and analysis are not deep enough. It’s unclear how much each proposed module contributes. The paper would benefit from more detailed ablation and efficiency studies.
3.	Since the task involves generative image synthesis, some form of human preference or perceptual realism test would strengthen the claims.

**Questions:**

1. Ablation completeness:
Can you provide quantitative ablation results isolating the effects of (a) the triplane representation, (b) the ray-guided conditioning, and (c) the epipolar attention? How much performance gain does each contribute individually?
2. Did you conduct any small-scale user study or human preference comparison? If not, could you report preliminary results or plan to include one in the final version?
3. Since VIGOR++ uses Google Street View imagery, how is licensing and privacy handled?

---

> ### Author Response · Authors · 2025-11-22
> **Response to Reviewer QDgj (Part 1/2)**
>
> We thank Reviewer-QDgj for the valuable comments. Please find our response below.
>
> > **Q1. The core architectural ideas—triplane representation (EG3D), ray-based sampling (MVDream, Zero123++), and epipolar-constrained attention (EpiDiff)—are largely from prior work.**
>
> Regarding the use of triplane representation, our contribution lies in effectively integrating the triplane into a conditional satellite to street-view diffusion framework, enabling geometry-aware and controllable generation. Unlike methods such as MVDream and Zero123++, which implicitly encode camera embeddings directly into the diffusion process, we propose Ray-Based Pixel Attention, which explicitly samples points along viewing rays using learnable offsets and weights (Eq. 5). This design ensures spatial coherence and enhances geometric alignment across views. As shown in Tab. 1, compared with EscherNet—a method that relies on camera embeddings for pose control—our approach achieves superior performance in maintaining inter-frame consistency and scene realism in large-scale scene generation.
>
> For epipolar-constrained attention, we are inspired by prior work. However, unlike methods such as EpiDiff that focus on pinhole cameras, we apply this module to panoramic images, where equirectangular projections render epipolar lines nonlinear and conventional epipolar formulations inapplicable. Moreover, multiview consistency in satellite-to-street-view synthesis remains relatively underexplored. A most recent work[A] uses Full Cross-Attention across all frames, but this introduces high computational cost (as shown in Fig. 7 of the main paper) and weak geometric constraints (as shown in Tab. 4 of the main paper). In contrast, our Panoramic Epipolar Attention explicitly encodes pose-aware epipolar geometry, enabling attention along geometrically valid paths and significantly improving multiview coherence and structural consistency.
>
> > **Q2. Can you provide quantitative ablation results isolating the effects of (a) the triplane representation, (b) the ray-guided conditioning, and \(c) the epipolar attention?**
>
> We conduct additional ablation experiments to better isolate the effects of each module.
>
> **(Q2.a) Triplane Representation.**\
> We perform an ablation on the VIGOR dataset comparing the triplane representation with a standard BEV representation:
> ||↓$P_{alex}$|↓DINO|↓SegAny|↑SSIM|↑PSNR|↑SD|↓Depth|
> |-|-|-|-|-|-|-|-|
> |BEV|0.3803|4.408|0.3549|0.4134|13.64|12.94|7.061|
> |Triplane|**0.3469**|**4.287**|**0.3471**|**0.4385**|**14.08**|**13.11**|**6.727**|
>
> These results demonstrate that the triplane representation consistently outperforms the BEV representation across all metrics. This occurs because a single BEV plane cannot capture vertical structures. As a result, different camera rays—each requiring distinct elevation angles to render buildings accurately—end up sampling the same features, leading to incorrect reconstructions. The triplane effectively overcomes this limitation.
>
> **(Q2.b) Ray-Guided Conditioning.**\
> To validate the effect of the ray-guided conditioning mechanism, we compare our method with a vanilla conditioning approach, which performs cross-attention between every 3D point feature in the triplane and each image pixel. The ablation results on VIGOR are as follows:
> |            | ↓$P_{alex}$ | ↓DINO | ↓SegAny | ↑SSIM | ↑PSNR | ↑SD | ↓Depth |
> |------------|--------|-------|---------|---------|---------|-------|----------|
> | Vanilla condition   |  0.5413 | 5.425 | 0.3911  | 0.3174  | 12.35   | 12.05 | 25.38    |
> | Ray-guided condition   |  **0.3469** | **4.287** | **0.3471**  | **0.4385**  | **14.08**   | **13.11** | **6.727**    |
>
> The ray-guided conditioning substantially improves performance. This is because vanilla conditioning lacks explicit geometric constraints, often resulting in geometric distortions, whereas ray-guided conditioning leverages Ray-Based Pixel Attention to dynamically sample points along viewing rays using learnable offsets (Eq. 5). This approach ensures spatial coherence and improves geometric alignment across views.
>
> **\(Q2.c) Epipolar Attention.**\
> To evaluate the effect of epipolar attention, we compare our method with variants without multi-view attention (w/o Epipolar-Att) and with vanilla multi-view attention (w/ Vanilla MV-Att) on the VIGOR++ dataset:
> | Method             | ↓DINO   | ↓SegAny  | ↑PSNR  | Depth↓  | ↓FVD   | ↓CLIPSIM |
> |-------------------|--------|---------|-------|--------|-------|---------|
> | w/o Epipolar-Att  | 4.275 | 0.3626  | **12.86** | 6.174  | 3.439 | 10.20   |
> | w/ Vanilla MV-Att | 4.253  | 0.3589  | 12.75 | 6.231  | 2.150 | 7.516   |
> | w/ Epipolar-Att   | **4.156**  | **0.3563**  | 12.75 | **5.623**  | **2.101** | **6.820**   |
>
> These results indicate that adding epipolar attention further improves performance, especially in terms of multi-view consistency, where the gain is substantial.

---

> ### Author Response · Authors · 2025-11-22
> **Response to Reviewer QDgj (Part 2/2)**
>
> > **Q3. Some form of human preference or perceptual realism test would strengthen the claims.**
>
> We thank the reviewer for this valuable suggestion. We conduct a user study, which we add in Appendix A.15. Specifically, we compute the Average User Ranking (AUR). We randomly sample 1,000 trajectories from the test set and collect rankings from five users to obtain the final AUR scores. Each user ranks the methods for each sequence: the top-ranked method receives 2 points, the second receives 1 point, and the third receives 0 points. The final AUR is computed by averaging the scores across users.
>
> The evaluation considers two aspects: similarity between the generated ground scenes and satellite images (S2G-Sim) and multi-view consistency (MV-Cons). The results are shown below:
> |       | ↑S2G-Sim | ↑MV-Cons |
> |------------|----------:|---------:|
> | EscherNet  |     0.481 |    0.974 |
> | ControlS2S |     1.057 |    0.051 |
> | Ours       |     **1.462** |    **1.975** |
>
> These results demonstrate that our method generates ground scenes that better reflect the satellite imagery and achieves superior multi-view consistency.
>
> > **Q4. How is the dataset licensing and privacy handled?**
>
> When releasing the dataset, to comply with Google’s terms of service and ensure proper privacy protection, we will do not release the raw Street View or satellite images themselves. Instead, following established practices in prior works such as VIGOR [B] and DReSS [C], we will release the panorama IDs, coordinates, and satellite zoom levels. This approach ensures full reproducibility of our work while respecting data usage policies.
>
> [A] Satellite to GroundScape - Large-scale Consistent Ground View Generation from Satellite Views, CVPR25\
> [B] VIGOR: Cross-View Image Geo-localization beyond One-to-one Retrieval, CVPR21\
> [C] Cross-View Geo-Localization with Panoramic Street-View and VHR Satellite Imagery in Decentrality Settings, JPRS25

---

### Official Review · Reviewer_7M3F · 2025-11-01

**Soundness:** 3
**Presentation:** 2
**Contribution:** 2
**Rating:** 6
**Confidence:** 3

**Summary:**

This paper introduces a diffusion-based framework capable of generating sequential ground-view images conditioned on a given satellite image and trajectory. The method employs a ray-based pixel attention mechanism to effectively aggregate features from a triplane representation. Furthermore, an Epipolar-Constrained Attention module is incorporated to enforce geometric consistency across multiple camera views. In addition, the authors release a new dataset establishing correspondences between satellite maps and ground-view sequences.

**Strengths:**

* The paper’s objectives are novel and clearly defined.
* The proposed approach is logically sound, and the contributions are substantial — including the introduction of a new dataset for a newly defined task.
* The experiment demonstrates the effectiveness of the proposed method with state-of-the-art performance.

**Weaknesses:**

* Some parts of the paper are not fully explained and require further clarification from the authors (Details shown in the Question part).
* The epipolar constraint only ensures local consistency between two frames rather than global consistency, which makes the overall constraint relatively weak.
* There aren’t enough experiments to examine how each module affects the consistency metric, which I personally consider a core metric for multi-image generation. The existing ablations—for example, comparing full cross-attention with epipolar attention—some consistency metrics (e.g., FVD), the enhancement does not seem significant. The authors need to include more comprehensive evaluations to substantiate their claim of enhancing consistency performance..

**Questions:**

1. What is the purpose of using a triplane representation? From the ablation studies (Table 5), it seems that the performance gain is not very significant even with increased computational resources. Moreover, since satellite images only provide 2D information, how can the features along the Z-axis in the triplane representation be effectively extracted without height information?

2. When constructing the dataset, how to determine the sampling density of ground images? Does the sampling density (high or low) affect the quality of the generated images?

3. How does the system deal with the dynamic object issue during the image generation process?

---

> ### Author Response · Authors · 2025-11-22
> **Response to Reviewer 7M3F (Part 1/3)**
>
> We thank Reviewer-7M3F for the valuable comments. Please find our response below.
>
> > **Q1. The epipolar constraint enforces only local consistency between two frames, making the overall constraint relatively weak.**
>
> We thank the reviewer for this comment. We agree that a pairwise epipolar constraint is local in nature, but our system is designed to leverage this local property to build strong global consistency. This is achieved through two key mechanisms in our framework:
>
> 1. The epipolar constraint is applied not just between a single pair. During the generation of each new frame, we enforce the constraint in two critical directions: with the very first frame of the sequence, and with the immediately preceding frame (t-1). The first frame acts as a global anchor, ensuring consistency in overarching scene characteristics such as weather and illumination. Simultaneously, the constraint with the previous frame preserves local geometric coherence. This dual application creates a transitive chaining effect, where geometric consistency propagates step-by-step through the sequence. As a result, local pairwise constraints collectively aggregate into a robust and globally consistent trajectory.
> 2. On the other hand, the locally consistent frames are all regularized by a global scene tri-plane representation. This representation acts as a unifying scaffold, ensuring that the geometry of each local image conforms to a single, coherent 3D structure across the entire scene.
>
> Furthermore, applying epipolar constraints to distant frames is intentionally avoided. In trajectory generation, frames far apart have little to no visual overlap, making the epipolar constraint geometrically ungrounded and computationally inefficient. Our focused application on nearby frames with sufficient overlap is a deliberate design choice that optimizes for both representational validity and memory usage.
>
> > **Q2. There are not enough experiments analyzing how each module affects the consistency metric and requests more comprehensive evaluations to support the claim of improved consistency.**
>
> To more thoroughly evaluate the contribution of the Epipolar-Constrained Attention mechanism to geometric consistency, we extend our ablation studies by removing this component and comparing it against a vanilla inter-frame attention baseline (Full Cross-Att).
>
> Beyond FVD, we additionally measure the average inter-frame similarity using a CLIP-based metric (CLIPSIM). We also conduct a user study, where we compute the Average User Ranking (AUR). Specifically, we randomly sample 1,000 trajectories from the test set and collect rankings from five users to obtain the final AUR scores. The results are summarized below:
>
> | Method            | ↓FVD  | ↓CLIPSIM | ↑AUR(seq) |
> |-------------------|------|---------|----------|
> | w/o Epipolar-Att  | 3.439 | 10.20  | 0.174    |
> | w/ Full Cross-Att | 2.150 | 7.516  | 1.136    |
> | w/ Epipolar-Att   | **2.101** | **6.820**  | **1.690**    |
>
> These results demonstrate that adding Epipolar-Constrained Attention yields a substantial improvement over the variant without it. Compared with the vanilla inter-frame attention, our method achieves a 10% gain in CLIPSIM, receives significantly higher user rankings, and, as shown in Fig. 7, substantially reduces computational resources. Together, these findings provide strong evidence that the proposed Epipolar-Constrained Attention effectively enhances inter-frame geometric consistency.

---

> ### Author Response · Authors · 2025-11-22
> **Response to Reviewer 7M3F (Part 2/3)**
>
> > **Q3： Triplane Representation Purpose and Z-Axis Feature Extraction.**
>
> We thank the reviewer for raising these important questions. The triplane representation is a pivotal component of our framework, specifically designed to overcome the inherent limitations of a pure Bird's-Eye View (BEV) representation for our task. Its purpose is to provide a computationally efficient yet richly structured 3D scene prior. A single BEV plane lacks any capacity to model vertical structure, meaning that distinct camera rays—which must account for different elevation angles to render buildings correctly—would incorrectly sample identical features. This fundamental lack of 3D support explains the consistent performance gap we report in Tab. 5, where the triplane representation significantly outperforms the BEV baseline.
>
> We also wish to underscore that this representational advantage is achieved with minimal computational penalty. The fused feature vector passed into the attention mechanism retains the same dimensionality as in the BEV case, and the number of samples per camera ray remains unchanged. The resulting memory overhead is modest, as evidenced by the reported difference of approximately 1.3 GB (batch size of 32), a cost that is clearly justified by the substantial gains in output quality and geometric consistency demonstrated in our experiments.
>
>
> Regarding the effective extraction of Z-axis features, it is crucial to clarify that the triplane learns this information implicitly through a fully differentiable rendering process. The three orthogonal feature planes are combined for any 3D point via summation. This aggregated feature is then utilized by our Ray-Based Pixel Attention module to generate the final output image. Because this pipeline is trained end-to-end, the supervision signal from the final generation loss—encompassing both photometric and adversarial objectives—directly propagates back through the network. This process forces the model to discover and encode meaningful Z-axis correlations, such as the geometry of building facades and the patterns of cast shadows, within the feature representations of the XZ and YZ planes.
>
>
>
> > **Q4. How to determine the sampling density of ground images? Does the sampling density affect the quality of the generated images?**
>
> In constructing the dataset, we aim to maximize the use of all available ground images within each region. We first build a connectivity graph by ensuring temporal and weather consistency, and then compute pairwise distances between ground images using their GPS coordinates. We retain only edges whose distances are within 20 m and whose yaw-angle differences are below 50°. Trajectories are then constructed based on this connectivity graph. As a result, the maximum interval between adjacent frames is 20 m, while the minimum interval is determined by the native spacing of the available ground images. Overall, the constructed dataset has an average frame interval of 11 m, with the smallest interval being 0.079 m and the largest being 20 m.
>
> To examine how sampling density affects generation quality, we reorganize the test set and evaluate trajectories under different frame-interval ranges. Specifically, we group trajectories into two categories: intervals <10 m and intervals ≥10 m. The evaluation results are shown below:
>
> | Interval | ↓$P_{alex}$ | ↓DINO | ↓SegAny | ↑SSIM  | ↑PSNR | ↓Depth | ↓CLIPSIM |
> |----------|---------|-------|---------|--------|-------|--------|-----------|
> | <10m     | **0.3611**  | **4.088** | **0.3420**  | **0.4592** | **13.35** | **4.314**  | **6.204**    |
> | ≥10m     | 0.3848  | 4.155 | 0.3530  | 0.4090 | 12.82 | 5.494  | 6.639     |
>
> As the frame interval increases, the generation task becomes more challenging because the correlation between adjacent frames becomes significantly weaker. Consequently, we observe a consistent trend of performance degradation at larger sampling distances.

---

> ### Author Response · Authors · 2025-11-22
> **Response to Reviewer 7M3F (Part 3/3)**
>
> > **Q5. Handling of Dynamic Objects.**
>
> We thank the reviewer for this insightful question. Modeling dynamic objects is a challenging and valuable aspect of scene generation.
>
> In the current work, our primary focus is to establish a robust baseline for the novel task of trajectory generation from a single satellite image. The core technical contribution lies in ensuring multi-view geometric consistency for the static scene structure—such as buildings, roads, and terrain—across generated ground-view sequences. This foundational consistency is the central problem we aim to solve, and as our results in Tab. 1 and Tab. 2 demonstrate, our method establishes a new state-of-the-art for this task.
>
> We fully acknowledge that handling dynamic objects like vehicles and pedestrians is a critical next step for achieving photorealism. We consider this a compelling and natural direction for future work. Our current framework, which learns a coherent 3D scene representation, provides a strong foundation upon which dynamic object layers and their motion models could be integrated. We appreciate the reviewer highlighting this important research pathway.

---

### Official Review · Reviewer_DioS · 2025-11-01

**Soundness:** 3
**Presentation:** 3
**Contribution:** 4
**Rating:** 8
**Confidence:** 4

**Summary:**

This paper introduces SatDreamer360, a generative framework capable of producing multi-frame geometrically consistent ground-level panoramic image sequences from a single satellite image and a predefined camera trajectory. For single-frame generation, the framework employs a Triplane representation and a ray-based pixel attention mechanism to enhance geometric fidelity and visual realism. To ensure spatial and temporal coherence across generated frames, the authors further propose a Panoramic Epipolar-Constrained Attention module, which explicitly models geometric relationships between different views. To evaluate the proposed one-to-many sequence generation task more comprehensively, the authors introduce the VIGOR++ dataset, which extends the original VIGOR dataset by incorporating additional cities and continuous ground-view trajectories, enabling a more complete assessment of the framework’s performance. Overall, the proposed panoramic sequence generation method significantly improves the geometric consistency and continuity of satellite-to-ground image synthesis, providing higher-quality data for downstream applications.

**Strengths:**

1. Introducing image sequence generation into the satellite-to-ground view image synthesis task enables geometrically consistent sequential images, which are more suitable for downstream applications such as autonomous driving and 3D reconstruction. This approach further enhances the practical value of cross-view generation.

2. The proposed ray-based pixel attention mechanism and the epipolar-constrained attention leverage the geometric priors inherent in the imaging process, thereby improving the interpretability of both the module and the overall framework.

3. The authors constructed the VIGOR++ dataset, which not only increases the diversity of data types by expanding the number of cities but also provides carefully processed street-view image trajectories. This dataset offers a comprehensive and reliable benchmark for the proposed one-to-many sequence generation task and holds great potential value in other fields such as urban perception, autonomous driving, and 3D reconstruction.

4. The authors conducted extensive and thorough experiments to validate their framework, reflected in the use of diverse evaluation metrics across multiple levels and detailed ablation studies for each module. The supplementary materials provide additional methodological details and experimental analyses, and the evaluation of the generated results in downstream tasks is particularly commendable.

**Weaknesses:**

1. The proposed sequence generation task relies on a single satellite image and predefined trajectory inputs. However, the authors provide insufficient introduction and discussion regarding the trajectory data, including details such as the number of frames within each trajectory, the spatial intervals between frames, and the relative positioning of the trajectories within the satellite imagery.

2. Although the proposed VIGOR++ dataset expands the number of cities to enhance data diversity, it remains limited to U.S. cities. The absence of data from non-U.S. regions such as Asia and Europe may pose potential limitations to the model’s generalization capability.

**Questions:**

1. It is recommended that the authors include a more detailed description of the dataset in the supplementary materials, such as the number of satellite and street-view images, the number of trajectories corresponding to each satellite image, and the distribution of the dataset across different cities.

2. Considering that the trajectory information is an essential component of the model input and directly governs the generated sequence outputs, I suggest that the authors further discuss the design of trajectory sequences. Specifically, how long the trajectories can be supported, how the sampling interval influences the generation results (as the current discussion only covers short sequences), and whether the relative spatial relationship between the trajectory and the satellite image affects the generation results.

3. Related to the previous point, the current experimental results mainly involve short and nearly straight trajectories. I am curious whether the proposed method can maintain strong geometric consistency when dealing with longer or curved trajectories, especially at intersections or turning points in urban environments.

4. The current visualizations effectively demonstrate the model’s strong multi-view geometric consistency. However, including some failure cases in the supplementary materials and analyzing the underlying reasons would make the framework more convincing.

5. In Figure 2, could the description of the EPIPOLAR-CONSTRAINED ATTENTION be made clearer? The explanation on the right side of this figure appears somewhat confusing.

6. A concern raised by the second limitation is whether the tri-plane representation can adequately replace traditional 3D representations in capturing more complex vertical spatial structures or non-planar terrains. Considering that the dataset primarily focuses on major U.S. cities with relatively flat terrain and similar urban layouts, it would be beneficial to include inference experiments on cities from Asia, Africa, or Europe, even if ground-truth references are not available.

---

> ### Author Response · Authors · 2025-11-22
> **Response to Reviewer DioS (Part 1/3)**
>
> We thank Reviewer-DioS for the valuable comments. Please find our response below.
>
> > **Q1. It is recommended to provide a more detailed dataset description, including image counts, trajectory, and city-level distribution.**
>
> We provide additional details about the dataset in App. A.6. The VIGOR++ dataset includes 91,498 satellite images and the same number of street-view images, as shown in Fig. 11(a). These images are evenly distributed across ten cities, covering a total area of 117.47 km².
>
> To construct trajectories, we first associate each ground-view image with its corresponding satellite patch, where the ground-view image lies at the center. Using this center point as the seed, we first filter out images captured under different weather conditions or on different dates, as these frames lack temporal consistency. We then construct a connectivity graph and apply depth-first search to extract all feasible routes. A semi-automatic procedure, combined with manual verification, then selects the longest and most temporally consistent route. Through this process, we obtain one trajectory for each satellite image.
>
> Overall, we generate over 90,000 cross-view satellite–ground sequence pairs. Among them, 84,055 pairs are used for training and 7,443 pairs are used for testing. As shown in Fig. 11(b), most trajectories consist of 7 to 16 ground-view frames. The average frame interval is approximately 11 m, with the smallest interval being 0.079 m and the largest reaching 20 m.
>
> > **Q2. Further analyze how trajectory sequences influence generation quality, including the trajectory length, the sampling interval, and the relative spatial relationship between the trajectory and the satellite image.**
>
> Thank you for pointing this out. We conduct additional experiments to analyze how trajectory design influences generation quality, and we include these analyses in App. A.13.
>
> **(Q2.a) Trajectory length.**\
> We reorganize the test set by grouping trajectories based on their total length into four ranges: <20 m, 20–40 m, 40–60 m, and >60 m. The results are shown below. The model performs well for trajectories shorter than 60 m, but we observe noticeable degradation when the trajectory length exceeds 60 m. This limitation primarily arises from our inability to train the model on sufficiently long sequences due to computational constraints. We expect this issue to be mitigated by incorporating longer trajectories during training in future work.
>
> | Distance | ↓$P_{alex}$ | ↓DINO | ↓SegAny | ↑SSIM  | ↑PSNR | ↓Depth | ↓CLIPSIM |
> |-----------|---------|-------|---------|--------|-------|--------|-----------|
> | <20m      | **0.3793**  | 4.182 | **0.3515**  | **0.4230** | **12.88** | 5.582  | 5.947     |
> | 20–40m    | 0.3812  | 4.159 | 0.3529  | 0.4141 | 12.79 | 5.422  | **5.921**     |
> | 40–60m    | 0.3850  | **4.143** | 0.3538  | 0.4072 | 12.75 | **5.417**  | 6.065     |
> | >60m      | 0.3972  | 4.160 | 0.3552  | 0.3839 | 12.69 | 5.760  | 6.744     |
>
> **(Q2.b) Sampling interval.**\
> To examine how sampling density affects generation quality, we reorganize the test set and evaluate trajectories under different frame-interval ranges. Specifically, we group trajectories into two categories: intervals <10 m and intervals ≥10 m. The evaluation results are shown below. As the frame interval increases, the generation task becomes more challenging because the correlation between adjacent frames becomes significantly weaker. Consequently, we observe a consistent trend of performance degradation at larger sampling distances.
>
> | Interval | ↓$P_{alex}$ | ↓DINO | ↓SegAny | ↑SSIM  | ↑PSNR | ↓Depth | ↓CLIPSIM |
> |----------|---------|-------|---------|--------|-------|--------|-----------|
> | <10m     | **0.3611**  | **4.088** | **0.3420**  | **0.4592** | **13.35** | **4.314**  | **6.204**    |
> | ≥10m     | 0.3848  | 4.155 | 0.3530  | 0.4090 | 12.82 | 5.494  | 6.639     |

---

> ### Author Response · Authors · 2025-11-22
> **Response to Reviewer DioS (Part 2/3)**
>
> **\(Q2.c) Relative spatial relationship between the trajectory and the satellite image**
>
> To evaluate how the relative spatial relationship between the trajectory and the satellite image affects generation quality, we shift the satellite image so that the trajectory appears at different locations within it. For example, when the satellite image is translated 20 m to the right, the trajectory—originally centered—becomes offset by 20 m. Using this setup, we test offsets of ±20 m and ±40 m. The results show that the generation quality is almost unaffected by these spatial shifts. This is primarily because our method does not naively use the entire satellite image as a global conditioning input. Instead, we extract geometry-aware features through Ray-Based Pixel Attention, which selectively samples informative points from the satellite representation along the ray directions. This sampling-based design effectively mitigates sensitivity to the global spatial alignment between the trajectory and the satellite image.
>
> | Offset | ↓$P_{alex}$ | ↓DINO | ↓SegAny | ↑SSIM  | ↑PSNR | ↓Depth | ↓CLIPSIM |
> |--------|---------|-------|---------|--------|-------|--------|-----------|
> | 0      | **0.3955**  | 4.156 | 0.3563  | **0.3964** | 12.75 | **5.623**  | 6.820     |
> | ±20    | **0.3955**  | **4.150** | **0.3560**  | 0.3962 | **12.78** | 5.631  | 6.745     |
> | ±40    | 0.3973  | 4.164 | 0.3567  | 0.3959 | 13.73 | 5.674  | **6.706**     |
>
>
> > **Q3. Whether the method can maintain strong geometric consistency on longer or curved trajectories, especially at intersections or turning points.**
>
> We thank the reviewer for this valuable point. In Fig. 17, we provide additional visualizations for long trajectories. The images are arranged from left to right and top to bottom, showing the generated results of a vehicle navigating a left-turn scenario. The entire trajectory spans approximately 70 meters, and the results demonstrate that our method has the ability to maintain high geometric consistency even for longer and curved paths.
>
> > **Q4. Analysis of failure cases.**
>
> Thanks for the valuable suggestion. We include additional failure cases in Fig. 12. Our method occasionally produces incorrect results when generating scenes in narrow alleyways. In particular, at the narrowest point of an alley (the third frame), the model sometimes interprets the scene as a tunnel. This issue arises because the dataset contains ground-view images captured along vehicle-accessible routes, and extremely narrow alleyways are generally not passable by vehicles. As a result, these scenarios are underrepresented in the training data, which leads to such failure cases.
>
>
> > **Q5. Could the description of the EPIPOLAR-CONSTRAINED ATTENTION be made clearer?**
>
> We thank the reviewer for the suggestion. In App. A.4, we provide a clearer explanation of the Epipolar-Constrained Attention mechanism and include additional visual illustrations.
>
> In Fig. 9, we show the full derivation of the panoramic epipolar geometry. Taking the camera positions $p^3$ and $p^1$ shown in Fig. 9(A) as an example, we consider the red point on the image plane of $p^3$ in Fig. 9(B). As illustrated in Fig. 9\(C), because $p^1$ uses an equirectangular projection, the ray corresponding to this point on $p^3$ is projected onto the image plane of $p^1$ as a green wavy curve. Since the back side of the sphere does not intersect with this ray, it can be omitted, resulting in the simplified curve shown in Fig. 9(D). Moreover, the valid portion of the ray from $p^3$ must lie in front of the camera, so points corresponding to negative ray directions can be removed. This yields the final green line segment in Fig. 9(E), which represents the valid epipolar region. The true correspondence of the red point on $p^3$ must lie on this green segment.
>
> By using this formulation, we no longer need to compare every point on the image plane of $p^3$ with all points on $p^1$. Instead, attention is restricted to the green line segment in Fig. 9(E). This significantly reduces computational complexity and avoids introducing noise through interactions with irrelevant points.

---

> ### Author Response · Authors · 2025-11-22
> **Response to Reviewer DioS (Part 3/3)**
>
> > **Q6.The inference experiments on cities from Asia, Africa, or Europe.**
>
> Thanks the reviewer for the suggestion. Training solely on U.S. cities and then performing inference on cities in Asia, Africa, or Europe poses a significant challenge due to the substantial differences in urban layouts, architectural styles, and street patterns across these regions. Under this setting, due to the difficulty of collecting ground-view images in Asia, we conduct testing using images from Europe and Africa. As shown in Fig. 15(Europe) and Fig. 16(Africa), other methods often fail under these conditions, whereas our approach consistently generates reasonable results, demonstrating robust generalization capabilities across diverse urban layouts. This performance is attributable to the tri-plane representation, which captures both horizontal and vertical features across multiple orthogonal planes, providing robust geometric support for various urban scenes. Furthermore, our Ray-Based Pixel Attention effectively aligns ground-view generation with the satellite representation, ensuring that the model maintains geometric consistency with satellite imagery across diverse city layouts.

---

> ### Comment · Reviewer_DioS · 2025-11-27
>
> I thank the authors for their detailed response and the efforts put into the revised manuscript. I have carefully reviewed the rebuttal materials and the updated paper, and I am glad to see that most of my previous technical concerns have been effectively addressed.
>
> However, before finalizing my recommendation, I have a few remaining inquiries and comments regarding the current version:
>
> 1. **Open Source Plan for VIGOR++:**
>    Considering the significant practical value and potential impact of **VIGOR++**, I am interested to know the authors' specific plans regarding open-sourcing the code and datasets. Clarifying this would greatly strengthen the contribution to the community.
>
> 2. **Visual Presentation Issues:**
>    I noticed some display defects in the revised PDF that hinder the reading experience. I strongly suggest the authors fix these in the final version:
>    * **Figure 7:** It appears to be not rendering correctly.
>    * **Figure 17:** This figure exhibits inconsistent resolutions compared to other figures. The figure appears blurrier than the rest, which makes it difficult to interpret the details.

---

> > ### Author Response · Authors · 2025-11-28
> >
> > Thank you for your great effort in helping us improve our paper! We are glad that most of your previous questions have been addressed.
> >
> > > **Q1. Open Source Plan for VIGOR++**
> >
> > We are committed to open-sourcing all code and dataset-related resources. Specifically, we will release the full training, inference, and evaluation codes, along with pretrained model weights. For the dataset, following the practices of [A] and [B], we will release the panorama IDs, coordinates, and satellite zoom levels necessary for reconstruction, which ensures full reproducibility of our work while respecting Google’s data usage policies.
> >
> > [A] VIGOR: Cross-View Image Geo-localization beyond One-to-one Retrieval, CVPR21\
> > [B] Cross-View Geo-Localization with Panoramic Street-View and VHR Satellite Imagery in Decentrality Settings, JPRS25
> >
> >
> > > **Q2. Visual Presentation Issues**
> >
> > We sincerely appreciate your careful observation. In the revised version, we have corrected all the visual issues:
> > * **Figure 7:** We observed that Figure 7 did not render correctly on certain devices (e.g., mobile platforms). This issue has now been fixed in the revised version, and the figure should display properly after clearing the browser cache.
> > * **Figure 17:** We have revised Figure 17 to make it consistent with other firgures and ensured that all visualizations follow a unified presentation style.
> >
> > Please let us know if you have any other suggestions to improve the quality of this paper. We are very happy to working on it. Your feedback is greatly appreciated!

---

### Meta-Review · Area_Chair_Cbde · 2025-12-05

**Summary:**

Across the four reviews, the paper is well-received, with **two strong accepts** (R1/DioS, R4/5Zje), **one weak accept** (R2/7M3F), and **one borderline reject (R3/QDgj)**. All reviewers agree the problem is important and underexplored, the proposed framework is technically sound, and the new dataset VIGOR++ is valuable.

The main concerns centered on
1. insufficient explanation of key design choices (triplane, ray-based conditioning, epipolar attention),
2. limited details on dataset construction and trajectory characteristics,
3. lack of comprehensive ablations and statistical significance tests, and
4. incomplete baseline comparisons. After the rebuttal, most concerns were addressed with substantial additional analyses, experiments, ablations, and clarifications.

Remaining issues are minor or conceptual and do not alter the overall positive assessment.

**Reviewer Concerns:**

**Addressed**

- Module justification + ablations (triplane vs BEV, ray-guided vs vanilla, epipolar vs full CA).
- Trajectory/dataset detail: frame intervals, lengths, sampling, construction process.
- Long/curved trajectory results and generalization (Europe/Africa).
- Human evaluation and multiple-seed statistical tests.
- Hyperparameter rationale (K, M, triplane resolution, batch sizes, LR).
- Baseline fairness: EscherNet adaptation explained; SVD partially evaluated.
- Open-sourcing plan and figure fixes.

**Partially Outstanding**

- Conceptual novelty concern (R3) remains partly subjective.
- Video diffusion baselines: only preliminary SVD results provided.
- Dynamic objects acknowledged as future work.
- No statistical variance for baseline metrics.

**Reviewer Scores:**

**Reviewer DioS (score 8): Satisfied; likely remains 8**

DioS stated that most concerns were effectively addressed, leaving only minor presentation and open-sourcing questions, both of which were resolved in the rebuttal.

**Reviewer 7M3F (score 6): Major concerns resolved; likely increases to 8 or remains 6**

7M3F’s major concerns were:

- Insufficient ablations
- Weak global consistency reasoning
- Need for more analysis

The rebuttal robustly addressed these with extensive new experiments.

**Reviewer QDgj (score 4): Most technical concerns resolved except novelty; likely increases to 6 or remains 4**

QDgj’s central concern was lack of novelty, which the rebuttal addressed partially but not fully (this was the most philosophical criticism). Nevertheless, the reviewer’s other technical concerns were fully resolved, and user study + module ablations were added.

**Reviewer 5Zje (score 8): Strongly satisfied; likely remains 8**

5Zje’s technical and methodological questions were thoroughly answered, including statistical significance, baselines, hyperparameters, and EscherNet adaptation.

---

### Decision · Program_Chairs · 2026-01-26

Accept (Poster)